# A Mettl16/m6A/*mybl2b*/Igf2bp1 axis ensures cell cycle progression of embryonic hematopoietic stem and progenitor cells

Yunqiao Han [1,2,8], Kui Sun [1,8], Shanshan Yu[3,8], Yayun Qin[4,8], Zuxiao Zhang[1], Jiong Luo[1], Hualei Hu[1], Liyan Dai [1], Manman Cui[5], Chaolin Jiang [2], Fei Liu[2], Yuwen Huang[1], Pan Gao[1], Xiang Chen[1], Tianqing Xin[6], Xiang Ren[1], Xiaoyan Wu[6], Jieping Song [4], Qing Wang [1], Zhaohui Tang[1], Jianjun Chen [7], Haojian Zhang [5], Xianqin Zhang [1✉], Mugen Liu [1✉] & Daji Luo [2✉]

## Abstract

**Prenatal lethality associated with mouse knockout of *Mettl16*, a recently identified RNA N6-methyladenosine (m6A) methyltransferase, has hampered characterization of the essential role of METTL16-mediated RNA m6A modification in early embryonic development. Here, using cross-species single-cell RNA sequencing analysis, we found that during early embryonic development, METTL16 is more highly expressed in vertebrate hematopoietic stem and progenitor cells (HSPCs) than other methyltransferases. In Mettl16-deficient zebrafish, proliferation capacity of embryonic HSPCs is compromised due to G1/S cell cycle arrest, an effect whose rescue requires Mettl16 with intact methyltransferase activity. We further identify the cell-cycle transcription factor *mybl2b* as a directly regulated by Mettl16-mediated m6A modification. Mettl16 deficiency resulted in the destabilization of *mybl2b* mRNA, likely due to lost binding by the m6A reader Igf2bp1 in vivo. Moreover, we found that the METTL16-m6A-*MYBL2*-IGF2BP1 axis controlling G1/S progression is conserved in humans. Collectively, our findings elucidate the critical function of METTL16-mediated m6A modification in HSPC cell cycle progression during early embryonic development.**

**Keywords** METTL16; Early Embryonic Development; Hematopoietic Stem and Progenitor Cells (HSPCs); *MYBL2*; Cell Cycle
**Subject Categories** Chromatin, Transcription & Genomics; Development; Haematology

## Introduction

Hematopoietic stem and progenitor cells (HSPCs) possess unique self-renewal capacity and multilineage differentiation potential, replenishing all blood lineages throughout life (Orkin and Zon, 2008). Precise cell cycle regulation of HSPCs is an essential aspect of normal development and adult homeostasis (Hao et al, 2016), dysregulation of which is frequently associated with an increased risk of hematopoiesis-related diseases, such as bone marrow failure syndrome (BMFS), myelodysplastic syndrome (MDS) and various hematologic malignancies (Al-Rahawan et al, 2008; Hao et al, 2016; Li et al, 2018a). Compared with the highly quiescent HSPCs in the bone marrow (BM) in adulthood, many of the HSPCs in the fetal liver (FL) during early embryonic development are actively proliferating (Pietras et al, 2011), which is also the basis of normal adult hematopoiesis in BM. Thus, it is critical to reveal the molecular mechanisms underlying the cell cycle progression of HSPCs in early embryonic development.

$N^6$-methyladenosine (m6A) is the most abundant internal mRNA modification in eukaryotes (Jia et al, 2013). It is mainly deposited by the m6A methyltransferase (writer) complex, composed of METTL3/METTL14 heterodimer and co-factors(Liu et al, 2014). The m6A demethylases (erasers), which include FTO and ALKBH5, can remove m6A methylation (Zhao et al, 2014; Zheng et al, 2013). The dynamic m6A modification of mRNAs can be recognized by different reader proteins (e.g., YTHDF1/2/3, YTHDC1/2, and IGF2BP1/2/3), leading to regulation of mRNA splicing, nuclear export, stability, and translation (Huang et al, 2018; Wang et al, 2015; Xiao et al, 2016). Emerging evidence indicates the importance of the m6A modification in adult HSPC biology. Ablation of *Mettl3* blocks adult HSPC differentiation and leads to the accumulation of HSPCs (Lee et al, 2019). The ablation of *mettl14* promotes adult HSPC differentiation (Weng et al, 2018).

[1]Key Laboratory of Molecular Biophysics of the Ministry of Education, College of Life Science and Technology, Huazhong University of Science and Technology, Wuhan, Hubei 430074, China. [2]Key Laboratory of Breeding Biotechnology and Sustainable Aquaculture, Institute of Hydrobiology, The Innovative Academy of Seed Design, Hubei Hongshan Laboratory, Chinese Academy of Sciences, Wuhan 430072, China. [3]Institute of Visual Neuroscience and Stem Cell Engineering, College of Life Sciences and Health, Wuhan University of Science and Technology, Wuhan, Hubei 430065, China. [4]Medical Genetics Center, Maternal and Child Health Hospital of Hubei Province, Wuhan, Hubei 430070, China. [5]Frontier Science Center for Immunology and Metabolism, Wuhan University, Wuhan 430071, China. [6]Department of Pediatrics, Union Hospital, Tongji Medical College, Huazhong University of Science and Technology, Wuhan, Hubei 430022, China. [7]Department of Systems Biology, Beckman Research Institute of City of Hope, Monrovia, CA 91016, USA. [8]These authors contributed equally: Yunqiao Han, Kui Sun, Shanshan Yu, Yayun Qin. ✉E-mail: xqzhang04@hust.edu.cn; lium@mail.hust.edu.cn; luodaji@ihb.ac.cn

Depletion of *Ythdf2* results in BM HSPC expansion (Li et al, 2018b; Mapperley et al, 2021; Paris et al, 2019). Nevertheless, the role of m⁶A in the development of embryonic HSPCs remains elusive.

METTL16 has been recently identified as a new m⁶A methyltransferase (Pendleton et al, 2017). Different from the conventional METTL3-METTL14 complex, METTL16 functions as a monomer (Doxtader et al, 2018; Mendel et al, 2018). Its recognition motif UACAGAGAA hairpin structure is also distinct from that of METTL3 and METTL14 (Mendel et al, 2018; Pendleton et al, 2017), suggesting that METTL16 has unique physiological functions and molecular mechanisms. However, due to the prenatal lethality of *Mettl16* knockout mice (Mendel et al, 2018), it is extremely difficult to comprehensively explore the physiological functions of METTL16 in early embryonic development.

Here, we show that *METTL16* is highly expressed in HSPCs among vertebrates in early embryonic development and is required for the maintenance of embryonic HSPCs in an m⁶A-dependent manner. Mettl16 stabilizes the expression of *mybl2b* mRNA to promote G1/S progression of HSPCs in vivo, which is mediated by m⁶A reader Igf2bp1. Our studies demonstrated the essential function of m⁶A in the cell cycle progression of HSPCs, and uncovered the m⁶A network of Mettl16 in HSPC cell cycle progression in vivo. Moreover, the METTL16-m⁶A-*MYBL2*-IGF2BP1 signaling axis in G1/S progression is conserved in humans.

# Results

## *METTL16* is highly expressed in embryonic HSPCs among vertebrates

To determine the role of m⁶A modification in the development of embryonic HSPCs, we first compared the expression level of a series of methyltransferases in HSPCs using three independent datasets of single-cell RNA sequencing obtained from humans, mice, and zebrafish in early embryonic development (Xia et al, 2021; Zheng et al, 2022). Strikingly, cross-species analysis revealed that the expression level of *METTL16* in HSPCs is significantly higher than most other m⁶A methyltransferases in the METTL family in humans, mice and zebrafish (Fig. 1A–F). The expression of *METTL16* is also evidently higher in HSPCs than downstream differentiated blood cells among vertebrates (Fig. 1A–F). These results imply that METTL16 may play an evolutionarily conservative key role in embryonic HSPCs among vertebrates.

## Mettl16 is intrinsically required for HSPC maintenance in vivo

To comprehensively investigate the functions of METTL16 in HSPCs in vivo, we generated a *mettl16⁻/⁻* zebrafish line using CRISPR/Cas9 technology (Appendix Fig. S1A–F) and tracked the behavior of HSPCs. The emergency of HSPCs in aorta-gonad-mesonephros (AGM) was normal at 28 h postfertilization (hpf) and 36 hpf (Fig. EV1A). Up to 60 hpf, HSPCs could correctly migrate to and seed in the caudal hematopoietic tissue (CHT) of *mettl16⁻/⁻* embryos (Figs. 2A,B,I and EV1A). Given the normal specification and localization of HSPCs in *mettl16⁻/⁻* embryos, we then

performed a time course analysis of *cmyb* expression from 3 days postfertilization (dpf) to 5 dpf. The population of HSPCs in the CHT was initially reduced in *mettl16* deficient embryos at 3 dpf (Fig. 2C,D,I), and the defect was much more severe at 4 dpf and 5 dpf (Fig. 2E–I). These data indicated that the maintenance of HSPCs was perturbed in the CHT of *mettl16⁻/⁻* embryos. To confirm this finding, we crossed *mettl16⁻/⁻* zebrafish with Tg (*cmyb*: EGFP), in which HSPCs could be visualized by EGFP expression. At 2 dpf, the number of EGFP-positive cells was normal in *mettl16⁻/⁻* embryos (Figs. 2J and EV1B), but gradually decreased from 3 to 5 dpf (Figs. 2J,K and EV1B). As a result, HSPC-derived definitive hematopoiesis was severely impaired in *mettl16* deficient embryos (Fig. 2L,M). Interestingly, primitive hematopoiesis (Appendix Fig. S2A) and committed erythromyeloid progenitors (EMPs) (Appendix Fig. S2B), which arise before the emergency of HSPCs, were unaffected in *mettl16* deficient embryos. Taken together, our results revealed that the maintenance of HSPCs in the CHT, but not their emergence or settlement, was impaired in *mettl16⁻/⁻* mutants.

The vascular niche in the CHT is essential for HSPC development, whose disruption could cause HSPC defects (Xue et al, 2017). Whole-mount in situ hybridization (WISH) and analysis of angiogenesis under a Tg (*flk1*: EGFP) transgenic background showed that the development of arteries and veins in the CHT appeared normal in *mettl16⁻/⁻* embryos (Fig. EV1C,D), indicating that the defects in HSPC development were not the results of abnormal vascular impairment. Accordingly, a multi-dimensional RNA-Seq (Xue et al, 2019) analysis confirmed the highest and the most continuous expression of *mettl16* in HSPCs compared with other cell types in CHT (Fig. EV1E,F), implying that Mettl16 safeguards HSPCs in a direct and cell-autonomous fashion.

We also examined the integrity of nonhematopoietic tissues by surveying the expression of associated marker genes. Development of the somite (*desma*), intestine (*ifabp*), and neuro (*huc*) showed no apparent changes in the absence of *mettl16* (Fig. EV1G). These results suggest that Mettl16 plays a specific role in HSPC maintenance.

## Mettl16 deficiency impairs HSPC expansion through G1/S cell cycle arrest

Since the CHT is a hematopoietic site for HSPC rapid expansion in zebrafish, similar to the FL in mammals (Murayama et al, 2006; Xue et al, 2017), we wondered whether the defective HSPC maintenance was triggered by blocked proliferation in *mettl16⁻/⁻* embryos. The number of proliferating HSPCs in the CHT of *mettl16⁻/⁻* mutants were comparable with siblings at 2 dpf (Fig. 3A). Nevertheless, proliferating HSPCs was decreased in *mettl16⁻/⁻* embryos at 3 dpf (Fig. 3A). The defects were more obvious at 4 dpf and 5 dpf (Fig. 3A). Consistently, the phospho-histone H3 (PH3) and proliferating cell nuclear antigen (PCNA) staining also demonstrated an evident decline of proliferating HSPCs in *mettl16⁻/⁻* embryos (Fig. 3B), implying that the impaired proliferation accounts for the HSPC phenotype in *mettl16* deficient embryos.

To rule out the possibility that the defect in HSPC maintenance was due to excessive differentiation or apoptosis, we examined the differentiation and apoptosis of HSPCs in *mettl16* deficient embryos. Compared to siblings, HSPC differentiation towards

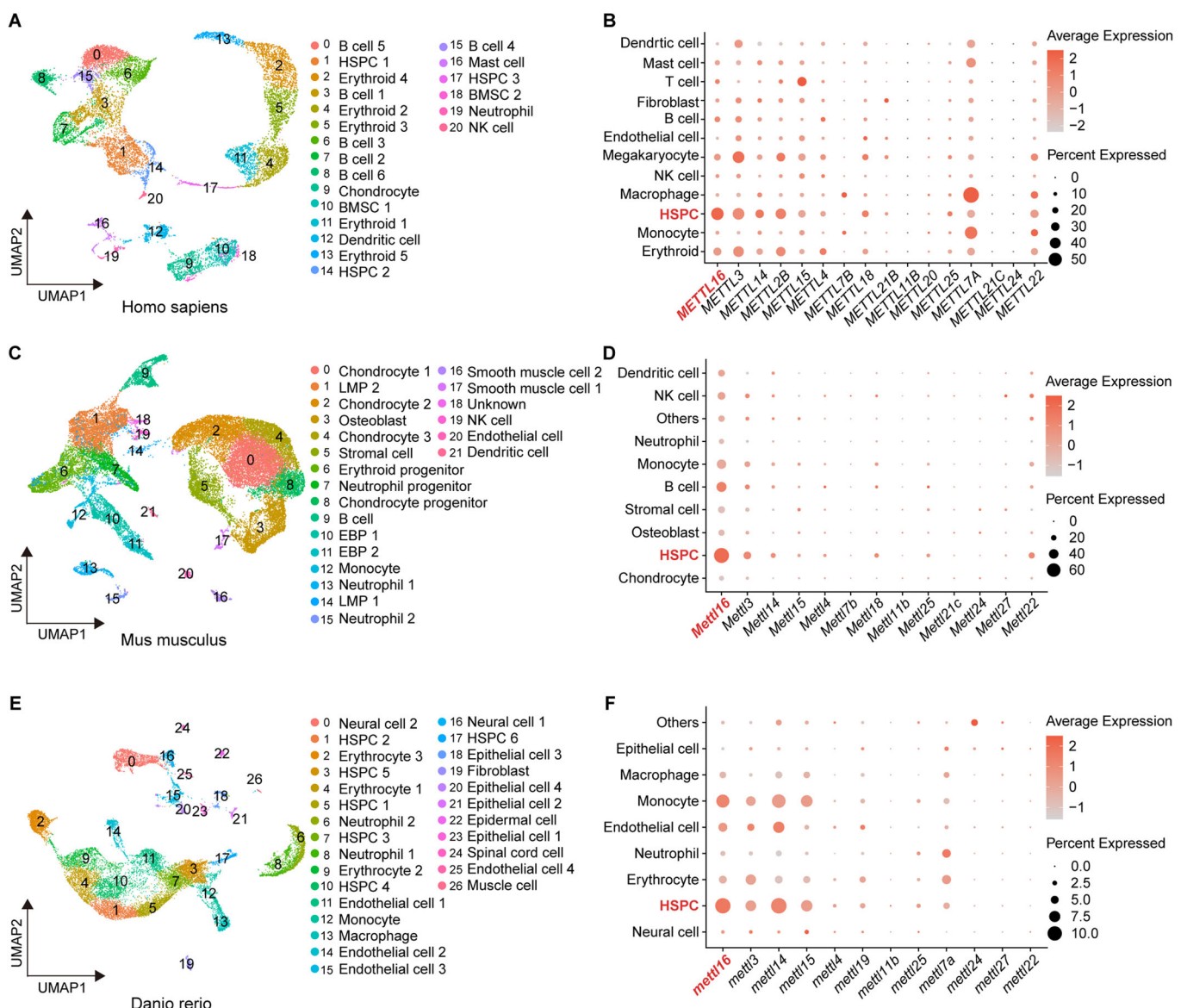

**Figure 1. METTL16 is highly expressed in HSPCs among vertebrates.**

(A) Uniform manifold approximation and project (UMAP) visualization of 21 hematopoietic cell clusters from the fetal liver of humans. HRA002414. BMSC bone marrow stromal cells. (B) Dot plot showing the expression of METTL genes in hematopoietic cell clusters in (A). (C) UMAP visualization of 22 hematopoietic cell clusters from the fetal liver of mice. CRA006858. LMP lymphoid-primed multipotent progenitors, EBP eosinophil/basophil progenitors. (D) Dot plot showing the expression of METTL genes in hematopoietic cell clusters in (C). (E) UMAP visualization of 27 hematopoietic cell clusters from the caudal hematopoietic tissue (CHT) of zebrafish. GSE146404. (F) Dot plot showing the expression of METTL genes in hematopoietic cell clusters in (E).

erythroid, myeloid, and lymphoid cells did not increase in *mettl16* deficient embryos (Fig. EV2A–C). Additionally, there was no discernible difference in apoptotic HSPCs in *mettl16*$^{-/-}$ embryos from 3 to 5 dpf (Fig. EV2D), supporting that the defective HSPC maintenance did not result from excessive differentiation or apoptosis in *mettl16*$^{-/-}$ embryos.

To further validate the impaired proliferation of HSPCs in *mettl16*$^{-/-}$ embryos, we used a *mettl16* splice morpholino (spMO) (Fig. EV3A), which reproduced the *mettl16* knockout effects on HSPCs. Consistently, *mettl16* deficiency reduced the number of HSPCs in the CHT from 3 dpf to 5 dpf (Fig. EV3B). As a result,

HSPC differentiation towards erythroid, myeloid, and lymphoid cells was nearly abolished in *mettl16* morphants at 5 dpf (Fig. EV3C). Similar to *mettl16* knockout zebrafish, proliferating HSPCs was also significantly reduced in *mettl16* morphants at 3 dpf and 4 dpf (Fig. EV3D), without excess apoptosis or differentiation (Fig. EV3E–G).

Normal cell cycle progression is crucial for cell proliferation (Liu et al, 2019). Therefore, we investigated whether there was cell cycle arrest of HSPCs in the CHT of *mettl16* morphants. Flow cytometry results showed that HSPCs were arrested in the G0/G1 phase in *mettl16* deficient zebrafish (Figs. 3C–E and EV3H). The protein

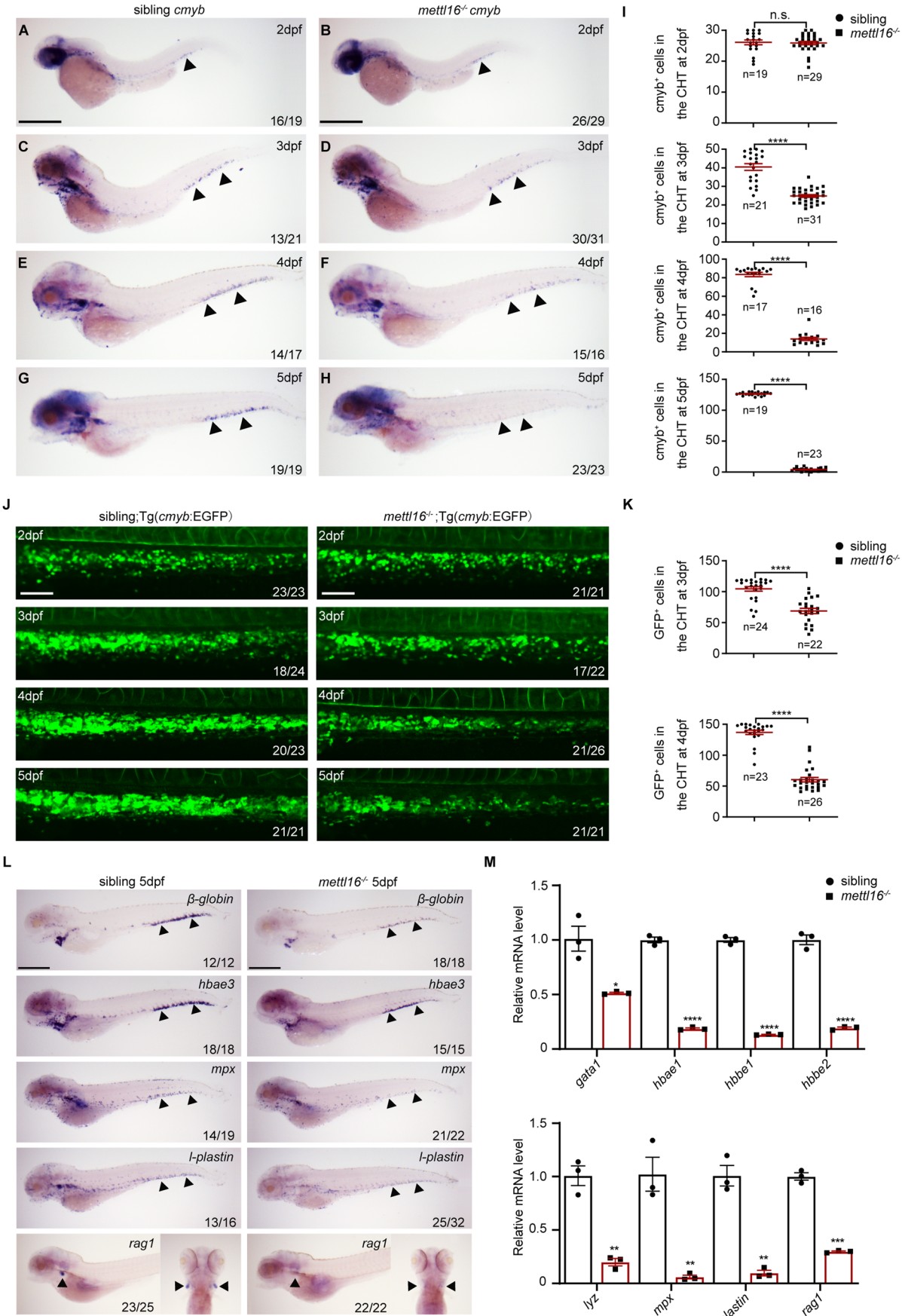

**Figure 2. HSPC deficiency initiates in the CHT of mettl16$^{-/-}$ embryos.**

(A–H) The whole-mount in situ hybridization (WISH) assay shows the expression of HSPC marker *cmyb* in the CHT of siblings and *mettl16* mutants from 2 to 5 dpf. Numbers at the bottom right indicate the number of embryos with similar staining patterns among all embryos examined. $n = 3$ independent experiments. The black arrowheads indicate *cmyb*$^+$ cells in the CHT. Scale bars, 100 μm. (I) Quantification of WISH in (A–H). '*n*' indicates the number of individuals analyzed for *cmyb*$^+$ cells in the CHT of siblings and *mettl16* mutants, respectively, from 2 to 5 dpf. (J) Live imaging showing the number of EGFP-positive HSPCs in the CHT region of siblings/Tg (*cmyb*: EGFP) and *mettl16*$^{-/-}$/Tg (*cmyb*: EGFP) zebrafish from 2 to 5 dpf. Numbers at the bottom right indicate the number of embryos with similar staining patterns among all embryos examined. $n = 3$ independent experiments. Scale bars, 20 μm. (K) Quantification of live imaging in (J). '*n*' indicates the number of individuals analyzed for GFP$^+$ cells in the CHT of siblings and *mettl16* mutants at 3 and 4 dpf, respectively. (L) Expression of differentiation markers β-*globin* (erythroid), *hbae3* (erythroid), *mpx* (myeloid), *l-plastin* (myeloid), and *rag1* (lymphoid) in sibling and *mettl16*$^{-/-}$ embryos at 5 dpf by WISH. Numbers at the bottom right indicate the number of embryos with similar staining patterns among all embryos examined. $n = 3$ independent experiments. The black arrowheads indicate differentiated blood cells. Scale bars, 100 μm. (M) qPCR analysis showing the expression of erythroid cell markers (upper) and myeloid cell markers (lower) in *mettl16* mutants at 5 dpf. $n \geq 15$ per group, performed with three biological replicates. Data information: In (I, K, M), data were represented as mean ± SEM. *adjusted $P < 0.05$, **adjusted $P < 0.01$, ***adjusted $P < 0.001$, ****adjusted $P < 0.0001$, n.s. non-significant, Student's *t*-test. Source data are available online for this figure.

expression level of P-RB, which promotes G1/S phase progression, was also significantly decreased in *mettl16* deficient zebrafish (Fig. EV3I), demonstrating that *mettl16* deficiency inhibited HSPC proliferation through G1/S cell cycle arrest.

DNA damage and the P53 signaling pathway are intensively linked to the events of cell cycle regulation (Engeland, 2018; Kastan and Bartek, 2004). Therefore, we defined whether *mettl16* deficiency hyperactivates DNA damage and the P53 signaling pathway. In *mettl16* deficient embryos, the protein levels of γ-H2AX and P-ATM (which mark DNA damage), RAD51 (which mediates repair of DNA double-strand breaks), and P53 were equal to that in siblings (Fig. EV2E), suggesting that there was no apparent DNA damage or activation of the P53 signaling pathway in *mettl16* deficient embryos. Moreover, the ablation of the P53 function failed to restore HSPC proliferation in *mettl16* deficient embryos (Fig. EV2F). These data indicated that DNA damage and the P53 signaling pathway did not contribute to the cell cycle arrest of HSPCs in *mettl16* deficient embryos.

## Disruption of Mettl16 reduced the m⁶A level of transcripts involved in the cell cycle process

Recently, METTL16 was reported to exert an m⁶A-independent role in hepatocellular carcinoma cells *in vitro* (Su et al, 2022), raising a question about whether METTL16 plays an m⁶A-dependent or m⁶A-independent role in the development of HSPCs in vivo. Therefore, we performed rescue experiments using a wild-type Mettl16 and two Mettl16 mutants, one with exclusive cytoplasmic localization via deleting its canonical nuclear localization signal (NLS) and the other with mutations of highly conserved residues in the substrate binding domain abrogating the methyltransferase activity (PP180/181AA) (Pendleton et al, 2017; Su et al, 2022) (Figs. 4A and EV4A). The overexpression of wild-type Mettl16, but not the del-NLS mutant or catalytic-dead mutant (PP180/181AA), restored the number of HSPCs in the CHT of *mettl16* deficient zebrafish (Fig. 4B–D), indicating that the regulation of HSPC development mediated by Mettl16 mainly relies on its methyltransferase activity.

Given that *mettl16* deficiency-mediated HSPC defects in vivo is mainly m⁶A-dependent, we detected m⁶A levels in *mettl16*$^{-/-}$ embryos, and found that the m⁶A levels were decreased in *mettl16* deficient embryos (Fig. 4E). We further profiled RNA m⁶A methylation in zebrafish embryos at 3 dpf using MeRIP-seq. Consistent with previous mammalian studies, m⁶A peaks in zebrafish are significantly enriched in the RGACH motif (R = G/A; H = A/C/U) (Fig. EV4B). Although detected in 1286 non-coding

RNAs (ncRNAs; 4.8%), m⁶A mostly occurs in mRNAs (95.2%) (Fig. EV4C,D). Around 5.5% methylated mRNAs were found to contain at least four peaks (Fig. EV4E), which is identical to that reported in mice (Meyer et al, 2012).

Compared to siblings, there are 1135 m⁶A peaks significantly decreased in *mettl16*$^{-/-}$ embryos (Fig. 4F; Dataset EV1), which mostly occurs in mRNAs (94.09%) (Fig. EV4F,G). Cell cycle processes are significantly enriched in the genes with declined m⁶A peaks (Fig. 4G), suggesting the role of Mettl16-mediated m⁶A modification in cell cycle progression. These declined m⁶A peaks are abundant in coding sequences (CDSs) and 3' untranslated regions (UTRs) (Fig. 4H), which might influence the stability of mRNA transcripts. Notably, motif scanning analysis revealed that 34.71% of these decreased m⁶A peaks have a UACAGAGAA highly similar motif, which could be identified as potential Mettl16 targets (Fig. 4I; Dataset EV2).

Given the importance of m⁶A in the regulation of mRNA transcription, splicing, and stability (Huang et al, 2018; Xiao et al, 2016), we performed transcriptome analysis to gain insights into the mechanisms of HSPC development modulated by Mettl16. We observed an obvious alteration of the transcriptome in *mettl16*$^{-/-}$ embryos (Fig. 4J). Gene Set Enrichment Analysis (GSEA) revealed that *mettl16*$^{-/-}$ embryos exhibited broad downregulation of genes associated with cell proliferation (Fig. 4K). Notably, the cell proliferation pathway is commonly detected by RNA-seq and MeRIP-seq, suggesting it is the main pathway enriched with potential Mettl16 targets. Additionally, we also found a degree of alterations in alternative splicing (Fig. EV4H).

## *mybl2b* is a novel functional essential target of Mettl16 in HSPCs

Through the integration of m⁶A-seq and RNA-seq data, we identified ten potential targets of Mettl16 in HSPCs, with eight being positively and two being negatively regulated by Mettl16 (Fig. 5A; Dataset EV3). Among them, the downregulation of *mybl2b*, *trpa1b*, and *ntrk2b* is tightly associated with G1/S arrest (Cojocaru et al, 2021; Frau et al, 2011; Yuan et al, 2018). Noticeably, the qPCR results confirmed that the mRNA expression level of *mybl2b*, but not *trpa1b* or *ntrk2b*, was decreased in both *mettl16*$^{-/-}$ embryos and *mettl16* morphants at 3 dpf and 4 dpf (Fig. 5C–F). Consistently, the mRNA expression of downstream targets of *mybl2b*, which are involved in G1/S progression, was also declined in *mettl16* deficient embryos at 3 dpf (Fig. EV5A). Consistent with the m⁶A-seq data (Fig. 5B), MeRIP-qPCR results verified the decrease of the m⁶A level of *mybl2b* mRNA (Fig. 5G).

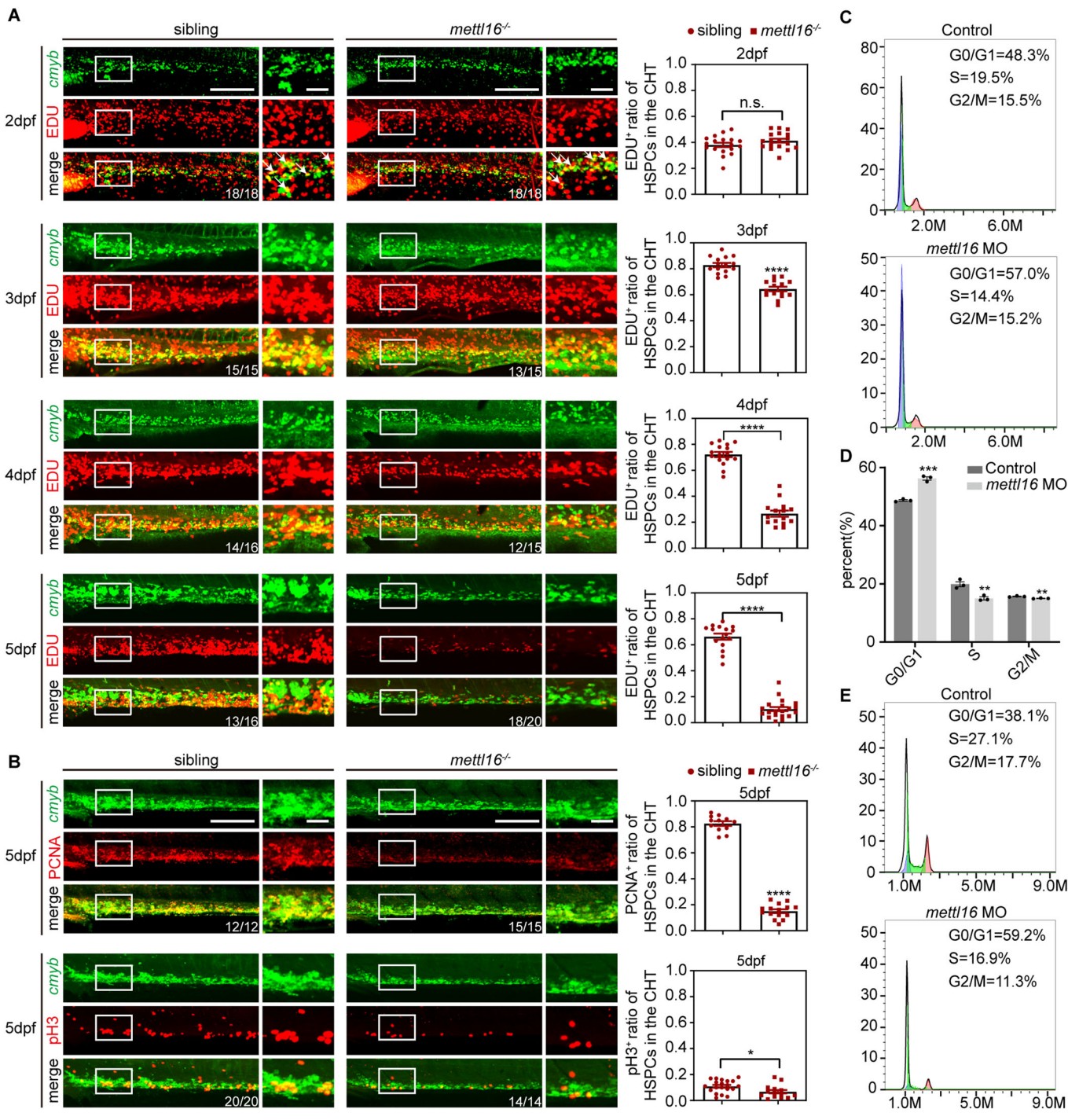

**Figure 3.  Depletion of Mettl16 inhibits HSPC proliferation through G1/S cell cycle arrest.**

(A) Double immunostaining of *cmyb*: EGFP and EDU showing the number of proliferating HSPCs in the CHT of siblings and *mettl16* mutants from 2 dpf to 5 dpf. Numbers at the bottom right indicate the number of embryos with similar staining patterns among all embryos examined. n = 3 independent experiments. The white arrowheads indicate proliferating HSPCs. Scale bars, 40 μm. (B) Double immunostaining of *cmyb*: EGFP and PCNA (upper panels) or pH3 (lower panels) showing the number of proliferating HSPCs in the CHT of siblings and *mettl16* mutants at 5 dpf. Numbers at the bottom right indicate the number of embryos with similar staining patterns among all embryos examined. n = 3 independent experiments. Scale bars, 40 μm. (C, D) Flow analysis showing the cell cycle of HSPCs of *mettl16*-deficient zebrafish at 3 dpf. n ≥ 200 per group, performed with three biological replicates. (E) Flow analysis showing the cell cycle of HSPCs of *mettl16*-deficient zebrafish at 5 dpf. n ≥ 200 per group, performed with three biological replicates. Data information: In (A, B, D), data were represented as mean ± SEM, * adjusted P < 0.05, ** adjusted P < 0.01, *** adjusted P < 0.001, **** adjusted P < 0.0001, n.s. non-significant, Student's *t*-test. Source data are available online for this figure.

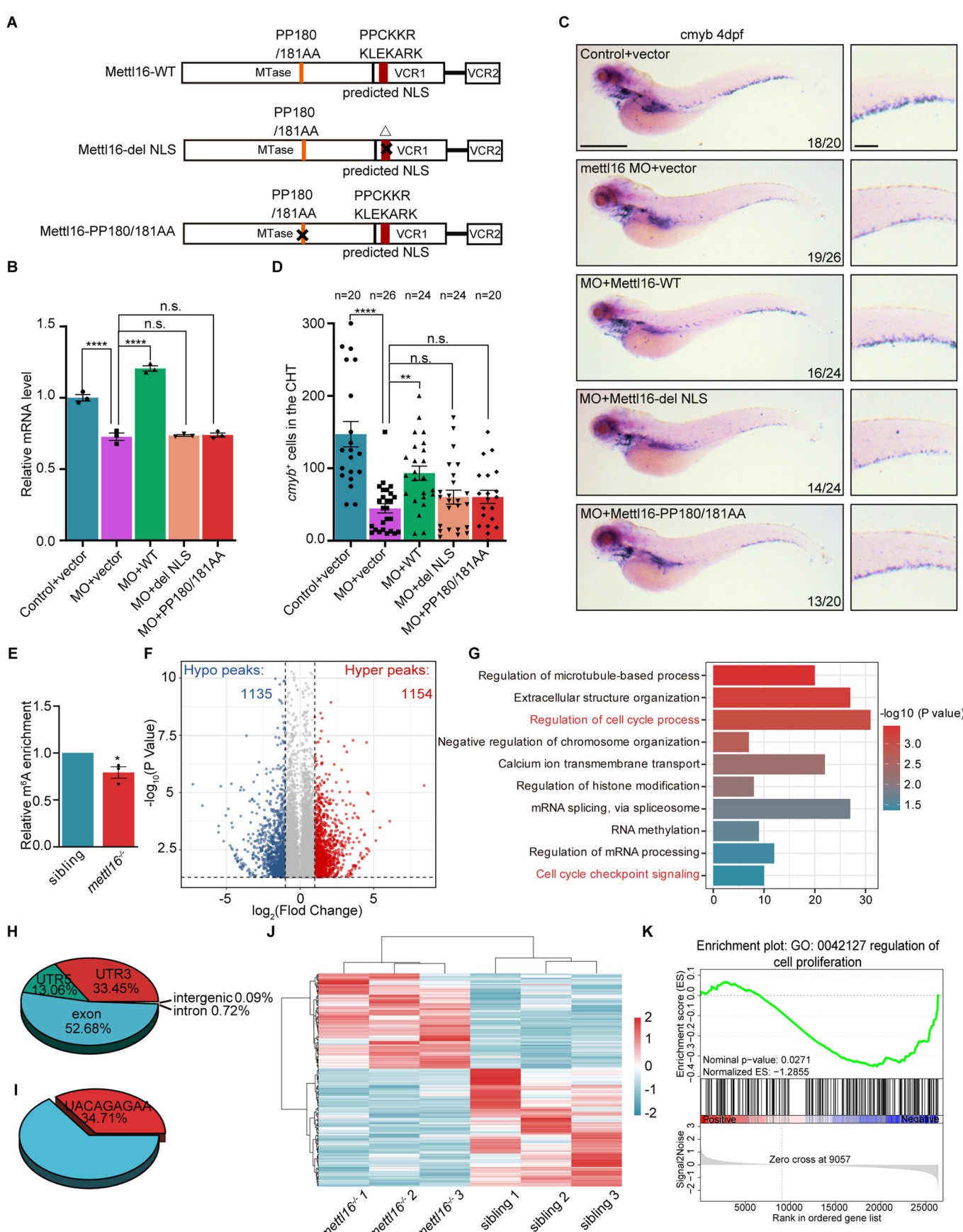

**Figure 4.   Disruption of Mettl16 invokes the alteration of m⁶A in cell cycle genes.**

(A) Schematic of Mettl16-WT and two Mettl16 mutants, including exclusive cytoplasmic Mettl16 (del-NLS) and catalytic-dead Mettl16 (PP180/181AA). (B, C) qPCR (B) and WISH (C) analysis showing the decreased expression of HSPC marker *cmyb* in *mettl16*-deficient zebrafish at 4 dpf was restored by Mettl16-WT, but not Mettl16-del-NLS and Mettl16-PP180/181AA. $n \geq 10$ per group, performed with three biological replicates (B). Numbers at the bottom right indicate the number of embryos with similar staining patterns among all embryos examined, $n = 3$ independent experiments (C). Scale bars, 100 μm (C). (D) Quantification of WISH in (C). 'n' indicates the number of individuals analyzed for *cmyb*⁺ cells in the CHT at 4 dpf, respectively. (E) m⁶A enrichment in RNAs in the CHT of siblings and *mettl16*-deficient zebrafish at 3 dpf. The m⁶A/A level of siblings was set to one. $n \geq 20$ per group, performed with three biological replicates. (F) Volcano plots reveal the differential m⁶A peaks of total RNAs in *mettl16*⁻/⁻ mutants compared with siblings at 3 dpf. The significantly increased and decreased m⁶A peaks are highlighted in red and blue, respectively ($P \leq 0.05$, |log₂FC| ≥1). Statistical analysis was performed by a hypergeometric test. (G) Gene ontology analysis of the m⁶A peaks that decrease upon *mettl16* knockout. Statistical analysis was performed by a hypergeometric test. (H) Pie chart depicting the annotations of the m⁶A peaks that decrease upon *mettl16* knockout. (I) Pie chart depicting the percent of the significantly decreased m⁶A peaks which have a UACAGAGAA highly similar motif upon *mettl16* knockout. (J) Heatmap showing differentially expressed genes in *mettl16*⁻/⁻ mutants compared with siblings at 3 dpf. (K) GSEA plot showing the general downregulation of genes involved in the regulation of cell proliferation in *mettl16*⁻/⁻ mutants compared with siblings. Statistical analysis was performed by permutation test using GSEA software. Data information: In (B, D, E), data were represented as mean ± SEM, *adjusted $P < 0.05$, ** adjusted $P < 0.01$, **** adjusted $P < 0.0001$, n.s. non-significant, one-way ANOVA analysis with post hoc test of Tukey's multiple comparison correction (B, D), Student's *t*-test (E). Source data are available online for this figure.

We then sought to determine whether *mybl2b* was a functionally essential target of Mettl16 in HSPC development in vivo. Remarkably, a multi-dimensional RNA-Seq (Xue et al, 2019) analysis revealed that the expression of *mybl2b* in HSPCs is higher and more durable than that in other cell types in CHT (Fig. EV5B). WISH also confirmed the high expression of *mybl2b* in the CHT (Fig. EV5C), indicating that *mybl2b* plays a vital role in HSPC development. Furthermore, our rescue assay showed that reconstitution of *mybl2b* largely restored the defect of HSPCs in the CHT of *mettl16* deficient embryos (Fig. 5H,I). G1/S arrest of HSPCs in the CHT was also alleviated in *mettl16* deficient embryos by *mybl2b* overexpression (Fig. 5J,K). These data strongly support *mybl2b* as a functionally essential m⁶A target of Mettl16 in HSPC development.

Considering that *U6* snRNA is a target of METTL16, which is a core element of splicing (Pendleton et al, 2017), we also conducted an integrative analysis of the m⁶A-seq and splicing-seq data. There are ten genes with both hypomethylated m⁶A peaks and aberrant splicing, of which five are related to G1/S arrest (Bai et al, 2018; Cao et al, 2018; Jiang et al, 2020; van de Wetering et al, 2002; Wang et al, 2021) (Fig. EV5D; Dataset EV4). However, there was no obvious aberrant splicing of these genes in *mettl16*⁻/⁻ embryos (Fig. EV5E).

METTL16 could promote the splicing or stability of *MAT2A* mRNA to maintain SAM homeostasis, and was proposed to modulate m⁶A deposition in an indirect manner (Pendleton et al, 2017). To investigate the possibility, we detected the splicing and expression of *mat2a* mRNA. Surprisingly, there was no discernible aberrant splicing or declined expression of *mat2a* mRNA in *mettl16*⁻/⁻ embryos at 3 dpf (Fig. EV5F,G), suggesting that *mybl2b* might be a direct m⁶A target of Mettl16. Motif analysis revealed that there is indeed a UACAGAAAA hairpin structured motif in the m⁶A peak of *mybl2b* mRNA (Fig. 6A), which is highly similar to the canonical m⁶A motif deposited by METTL16 (Pendleton et al, 2017). Furthermore, RIP-qPCR results demonstrated that Mettl16 could directly bind *mybl2b* mRNA (Fig. 6B), supporting that *mybl2b* mRNA is a direct m⁶A target of Mettl16.

To test whether Mettl16 regulates *mybl2b* expression through mRNA decay, we measured the *mybl2b* mRNA level in embryos after treatment with the transcription inhibitor actinomycin D. Compared to controls, the remaining *mybl2b* mRNA levels were significantly lower in *mettl16* deficient embryos, owing to the accelerated *mybl2b* mRNA decay in the absence of functional m⁶A modification in *mettl16* deficient embryos (Fig. 6C).

## The METTL16/m⁶A/*MYBL2*/IGF2BP1 axis in G1/S progression is conserved in zebrafish and humans

The effect of m⁶A modification on mRNA is mediated by specific m⁶A binding proteins known as m⁶A readers (Huang et al, 2018; Patil et al, 2018). Therefore, we investigate which reader mediates the effect of Mettl16 on the *mybl2b* mRNA stability in vivo. We checked all published RIP-seq and RNA-seq datasets of m⁶A readers using the RM2Target database (Bao et al, 2022), and found that *MYBL2* (human homology of *mybl2b* in zebrafish) mRNA could be bound by IGF2BP1 and IGF2BP3 (Fig. EV5H), which can promote the stability of their m⁶A targets (Huang et al, 2018; Wan et al, 2022; Xue et al, 2021). The expression of *MYBL2* was steadily decreased only in *IGF2BP1* depleted cells, but not in *IGF2BP3* KD or KO cells (Fig. EV5I), indicating that IGF2BP1, but not IGF2BP3, likely regulates the expression of *MYBL2*. Along with this, *IGF2BP1* knockdown could lead to G1/S arrest by destabilizing a panel of mRNAs (Muller et al, 2020; Sperling et al, 2022). Therefore, we consider Igf2bp1 as a potential m⁶A reader of *mybl2b*, mediating the effect of Mettl16 on *mybl2b* mRNA stability via m⁶A modification. RIP-qPCR results demonstrated that Igf2bp1 could bind *mybl2b* mRNA directly (Fig. 6D). Collectively, these results suggested that *mybl2b* mRNA is a direct m⁶A target of Mettl16, and Mettl16 regulates mRNA stability of *mybl2b* through m⁶A reader protein Igf2bp1.

Since the METTL16 protein is conserved in vertebrates (Pendleton et al, 2017), we sought to determine whether the METTL16-m⁶A-*MYBL2*-IGF2BP1 signaling axis in G1/S progression is conserved in humans. Our results showed that knockdown of *METTL16* triggered G1/S arrest in K562 cells (Fig. 6E–H). Consistent with *mettl16* deficient zebrafish, the m⁶A level of *MYBL2* was also evidently decreased upon *METTL16* depletion (Figs. 6I and EV5J). Consequently, the RNA expression level of *MYBL2* was declined due to strengthened mRNA decay (Fig. 6J, K). RIP-qPCR results demonstrated that METTL16 could directly bind *MYBL2* mRNA (Fig. 6L, M), suggesting that *MYBL2* is a direct m⁶A target of METTL16. Similar to that in zebrafish, m⁶A reader protein IGF2BP1 could directly bind *MYBL2* mRNA (Fig. 6N, O). Knockdown of *IGF2BP1* also caused the decreased mRNA stability of *MYBL2* (Fig. 6P–S). Taken together, our findings support a well-conserved role of the METTL16-m⁶A-*MYBL2*-IGF2BP1 signaling axis in G1/S progression in vertebrates. To further verify the conservation of the mechanism in the cell cycle progression of

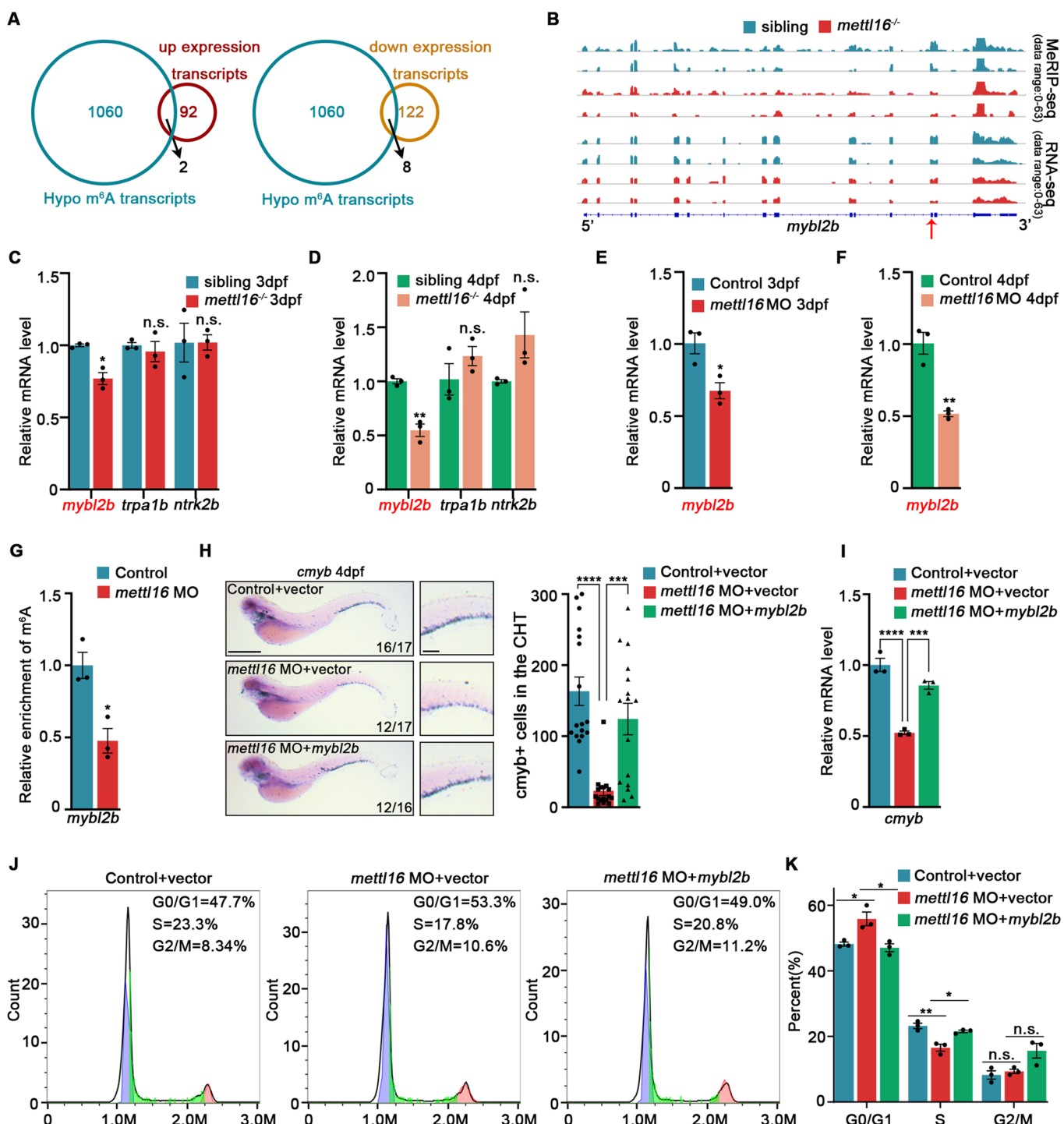

human HSPCs, we isolated CD34+ cells from human cord blood and knocked down *METTL16* (Fig. EV6A–C). The human cord blood CD34+ HSPCs were arrested in the G0/G1 phase upon *METTL16* depletion (Fig. EV6D). Similar to that in zebrafish, the m6A level of *MYBL2* was also significantly decreased, which led to the declined mRNA expression level of *MYBL2* due to defective mRNA stability (Fig. EV6E–G). Collectively, our findings shed light on the crucial role of METTL16 in early embryonic development

and the METTL16-m6A-*MYBL2* signaling axis in HSPC cell cycle progression in vivo.

## Discussion

In this study, we demonstrated that compared with other methyltransferases, *METTL16* is highly expressed in embryonic

**Figure 5.  *mybl2b* is a functionally essential target of Mettl16 in HSPC development.**

(A) Venn diagram showing the overlap between hypomethylated transcripts and differentially expressed genes in *mettl16*[−/−] mutants compared with siblings. (B) Integrative Genomics Viewer (IGV) tracks displaying MeRIP-seq (upper panels) and RNA-seq (lower panels) read distribution in *mybl2b* mRNA of siblings and *mettl16*[−/−] mutants. The red arrow at the bottom of the tracks indicates the position of the m6A peak. (C, D) qPCR analysis showing that the mRNA expression level of *mybl2b*, *trpa1b*, and *ntrk2b* in *mettl16*[−/−] embryos at 3 dpf and 4 dpf. $n \geq 15$ per group, performed with three biological replicates. (E, F) qPCR analysis showed the mRNA expression level of *mybl2b* in *mettl16* morphants at 3 dpf and 4 dpf. $n \geq 15$ per group, performed with three biological replicates. (G) m6A enrichment in *mybl2b* mRNA in *mettl16* deficient zebrafish at 3 dpf by meRIP-qPCR. $n \geq 200$ per group, performed with three biological replicates. (H, I) WISH (H) and qPCR (I) analysis showed that the decreased expression of HSPC marker *cmyb* in *mettl16* morphants at 4 dpf was restored by *mybl2b* mRNA co-injection. Numbers at the bottom right indicate the number of embryos with similar staining patterns among all embryos examined, $n = 3$ independent experiments (H). Scale bars, 100 µm (H). $n \geq 10$ per group, performed with three biological replicates (I). (J, K) Flow analysis of the cell cycle showing the increase of HSPCs in the G0/G1 phase in *mettl16* morphants was restored by *mybl2b* mRNA co-injection. $n \geq 200$ per group, performed with three biological replicates. Data information: In (C–I, K), data were represented as mean ± SEM, *adjusted $P < 0.05$, ** adjusted $P < 0.01$, *** adjusted $P < 0.001$, **** adjusted $P < 0.0001$, n.s. non-significant, Student's *t*-test (C–G), one-way ANOVA analysis with post hoc test of Tukey's multiple comparison correction (H, I, K). Source data are available online for this figure.

HSPCs among vertebrates, indicating the evolutionarily conservative role of *METTL16* in HSPCs in early embryonic development. The deficiency of Mettl16 specifically impaired the proliferation of embryonic HSPCs due to G1/S arrest. Despite the conditional deletion of *Mettl3* impaired FL HSPC expansion in murine and Mettl3 is required for HSPC specification in zebrafish (Gao et al, 2020; Zhang et al, 2017), the role of other m6A factors in early embryonic HSPC development is still unclear. Our results showed that the Mettl16 is specifically required for embryonic HSPC cell cycle progression, consistent with its high expression in HSPCs in early embryonic development. Though METTL16 was reported to influence erythropoiesis and leukemogenesis (Han et al, 2023; Yoshinaga et al, 2022), its role in HSPC cell cycle progression has never been elucidated. Our data, for the first time, revealed a critical role of Mettl16 in HSPC cell cycle progression.

In *mettl3* deficient zebrafish embryos, the emergency of HSPCs is blocked at 28 hpf. While *mettl16* depletion does not impair the emergency of HSPCs, but causes the defect of HSPC maintenance at 3 dpf. These results demonstrated the importance of m6A modification in embryonic HSPC development. And different m6A regulators may play specific roles in the different stages of HSPC development.

METTL16 has been reported to have a function independent of m6A in liver cancer cells *in vitro* (Su et al, 2022); nevertheless, it is unclear whether METTL16 plays an m6A-dependent or m6A-independent role in vivo. Due to the prenatal lethality of *Mettl16* knockout mice (Mendel et al, 2018), it is extremely difficult to address this issue in mice. Here we showed that the over-expression of wild-type Mettl16, but not the del-NLS mutant or catalytic-dead mutant (PP180/181AA), could restore the number of HSPCs in *mettl16* deficient zebrafish. Our data provides clear evidence for the m6A-dependent role of Mettl16 in HSPC expansion in vivo.

METTL16 is a new m6A methyltransferase identified recently, which adds m6A modification to RNA in the nucleus (Mendel et al, 2018; Pendleton et al, 2017; Warda et al, 2017). Yet to date, only a handful of m6A substrates of METTL16 have been reported in vitro. We identified *mybl2b* mRNA as a direct m6A target of Mettl16 in vivo, which is involved in the cell cycle progression of HSPCs. We further found that the splicing and expression of *mat2a* mRNA were identical in *mettl16*[−/−] embryos compared with siblings. Additionally, our meRIP-seq data demonstrated that 34.71% of downregulated m6A peaks have a UACAGAGAA highly similar motif in *mettl16*[−/−] embryos, indicating that there might be more direct m6A targets of Mettl16. Despite the m6A peaks of 1060 transcripts were significantly decreased, there are only

214 differentially expressed genes in *mettl16*[−/−] embryos, suggesting the m6A modification deposited by Mettl16 on its targets may also play other important roles in RNA regulation, such as influencing the nuclear exportation, cytosolic localization and translation of RNA, which warrants further investigation in the future.

The effect of m6A modification on RNAs is mediated by specific m6A binding proteins known as m6A readers (Huang et al, 2018; Patil et al, 2018). Nevertheless, it is still unclear which reader mediated the impact of METTL16 on its m6A targets in vivo. Our results revealed that Igf2bp1, a well-known m6A 'reader', which regulates mRNA stability (Chen et al, 2020; Huang et al, 2018; Xue et al, 2021), could bind *mybl2b* mRNA, suggesting it plays a vital role in recognition of m6A modification on *mybl2b* mRNA. It is worth noting that the knockdown of *IGF2BP1* triggered G1/S arrest (Muller et al, 2020; Sperling et al, 2022) and the decline of *MYBL2* mRNA stability, which is consistent with the phenotype observed in *mettl16* deficient embryos. Our results highlighted the Mettl16-*mybl2b*-Igf2bp1 signaling axis in the cell cycle progression of HSPCs *in vivo*. Moreover, the METTL16-*MYBL2*-IGF2BP1 axis in G1/S progression is conserved in humans.

Dysregulation of cell cycle progression in HSPCs is the cause of many hematologic diseases, such as BMFS, MDS, and various hematologic malignancies (Al-Rahawan et al, 2008; Hao et al, 2016; Li et al, 2018a). Our study revealed that Mettl16 is an essential cell cycle modulator of HSPCs, suggesting the great therapeutic potential to target METTL16 in the treatment of these hematopoiesis disorders. Notably, the most common clinical feature of BMFS is ineffective hematopoiesis in multiple cell lineages resulting from the impaired HSPC expansion (Tsai and Lindsley, 2020; Zhang et al, 2016a), which is similar to the phenotype observed in *mettl16*[−/−] zebrafish. Importantly, the advantages of in vitro fertilization and the rapid development of zebrafish embryos (de Jong and Zon, 2005) make it possible for us to systematically examine the effects of Mettl16 inhibition on other types of tissues besides the hematopoietic system. Our results demonstrated that the integrity of vasculature, somite, intestine, and neuro in *mettl16* deficient embryos was equal to that in siblings, uncovering a specific role of Mettl16 in HSPC cell cycle progression. These data imply that the *mettl16*[−/−] zebrafish might be a good model of BMFS to conduct drug screens targeting Mettl16 or *mybl2b*.

In summary, our results, for the first time, elucidated the role of Mettl16 in embryonic HSPC cell cycle progression as an m6A writer, and established a m6A regulatory network of Mettl16 on HSPC cell cycle progression in vivo, which may lead to new strategies of therapeutic interventions for hematopoiesis disorders.

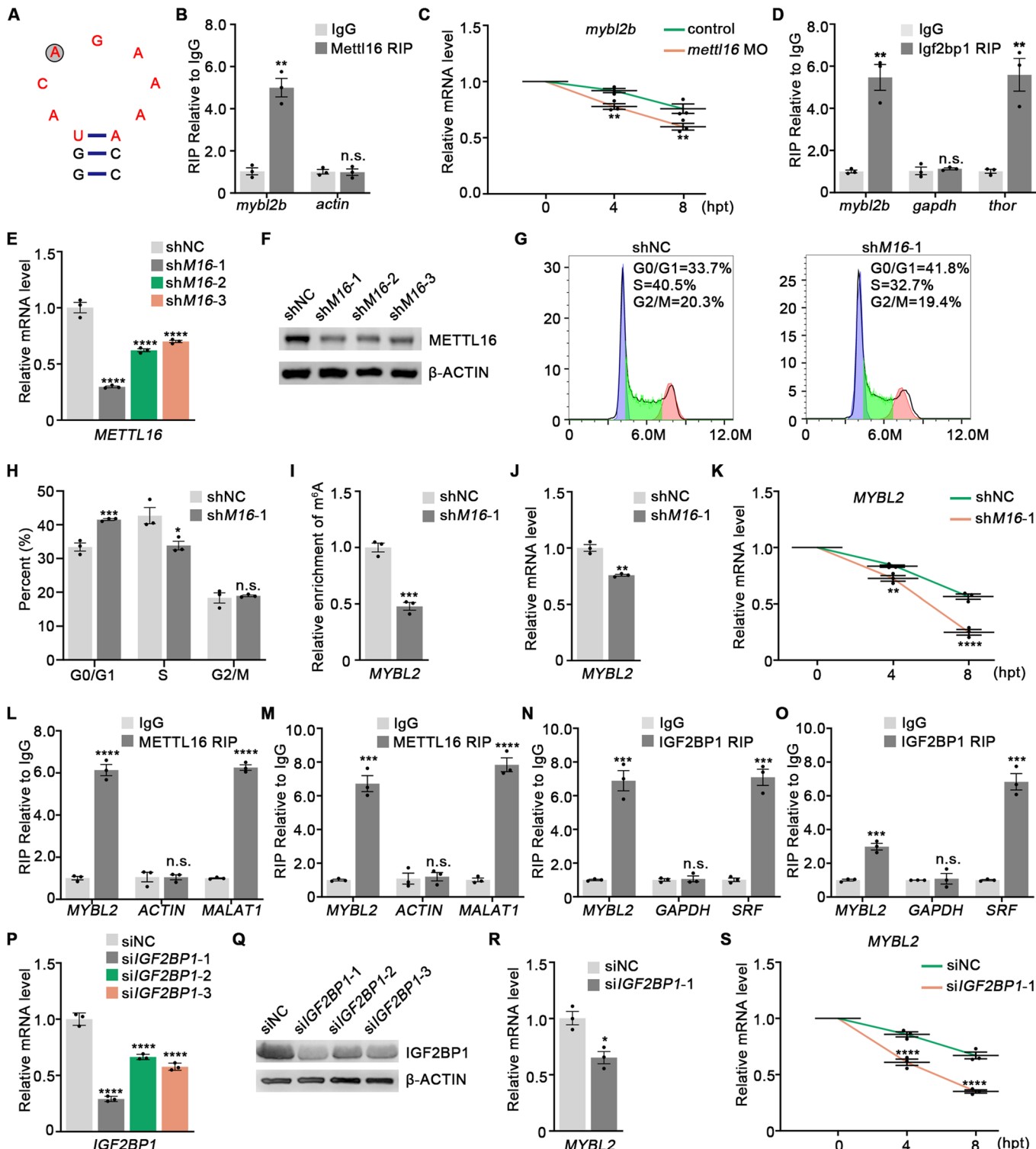

## Methods

### Zebrafish

Zebrafish of the AB strain were reared in a circulating water system at 28.5 °C and guaranteed on a 14-h-light/10-h-darkness cycle. Wild-type and injected embryos were maintained in E3 medium (5 mM NaCl,0.17 mM KCl, 0.33 mM CaCl$_2$, and 0.33 mM MgSO$_4$). If needed, 0.003% 1-phenyl-2-thiourea (PTU) (Sigma) was added at 13 hpf to suppress the pigmentation of embryos. The transgenic zebrafish used in this study, including Tg(*cmyb*:eGFP) and Tg(*flk*:mCherry), were purchased from the China Zebrafish Resource Center. All animal experimental procedures were reviewed and approved by the Ethics Committee of Huazhong

**Figure 6.   Mettl16 regulates mRNA stability of *mybl2b* through m⁶A reader protein Igf2bp1.**

(A) Predicted secondary structures surrounding m⁶A peak in *mybl2b* mRNA. The UACAGAAAAA box was shown in red and the m⁶A modification site was shown in gray circle. (B) Native RIP of Mettl16 with *mybl2b* mRNA in wild-type zebrafish. $n \geq 200$ per group, performed with three biological replicates. *actin* served as negative control. (C) qPCR analysis of embryos treated with actinomycin D for 4 and 8 h showed accelerated *mybl2b* mRNA degradation in *mettl16* morphants compared to control. $n \geq 20$ per group, performed with three biological replicates. (D) Native RIP of Igf2bp1 with *mybl2b* mRNA in wild-type zebrafish. $n \geq 200$ per group, performed with three biological replicates. *gapdh* and *thor* were served as negative and positive control, respectively. (E, F) Validation of the knockdown (KD) efficiency of the shRNAs against *METTL16* by qPCR (E) and western blot (F). $n = 3$ biological replicates. (G, H) Flow analysis showing increased G0/G1 phase in *METTL16* knockdown K562 cells. $n = 3$ biological replicates. (I) m⁶A enrichment in *MYBL2* mRNA in *METTL16* knockdown K562 cells by meRIP-qPCR. $n = 3$ biological replicates. (J) qPCR analysis showing the mRNA expression level of *MYBL2* in *METTL16* knockdown K562 cells. $n = 3$ biological replicates. (K) qPCR analysis of K562 cells treated with actinomycin D for 4 and 8 h showing accelerated *MYBL2* mRNA degradation in *METTL16* knockdown K562 cells. $n = 3$ biological replicates. (L, M) Native RIP of METTL16 with *MYBL2* mRNA in K562 (L) and HEK293 cells (M). $n = 3$ biological replicates. *ACTIN* and *MALAT1* were served as negative and positive control, respectively. (N, O) Native RIP of IGF2BP1 with *MYBL2* mRNA in K562 (N) and HEK293 cells (O). $n = 3$ biological replicates. *GAPDH* and *SRF* were served as negative and positive control, respectively. (P, Q) Validation of the knockdown (KD) efficiency of the siRNAs against *IGF2BP1* via qPCR (P) and western blot (Q). $n = 3$ biological replicates. (R) qPCR analysis showing the mRNA expression level of *MYBL2* in *IGF2BP1* knockdown HEK293 cells. $n = 3$ biological replicates. (S) qPCR analysis of HEK293 cells treated with actinomycin D for 4 and 8 h showing accelerated *MYBL2* mRNA degradation in *IGF2BP1* knockdown HEK293 cells. $n = 3$ biological replicates. Data information: In (B–E, H–P, R, S), data were represented as mean ± SEM, *adjusted $P < 0.05$, ** adjusted $P < 0.01$, *** adjusted $P < 0.001$, **** adjusted $P < 0.0001$, n.s. non-significant, Student's *t*-test (B, D, H–J, L, M, N, O, R), two-way ANOVA analysis with post hoc test of Tukey's multiple comparison correction (C, K, S), one-way ANOVA analysis with post hoc test of Tukey's multiple comparison correction (E, P). Source data are available online for this figure.

University of Science and Technology. All experiments comply with ethical regulations.

## Generation of *mettl16* mutants by CRISPR/Cas9

*mettl16⁻ᐟ⁻* zebrafish were generated using CRISPR/Cas9 technology as described previously (Li et al, 2021; Tu et al, 2022). To generate site-specific mutations in zebrafish, guide RNAs (gRNAs) were designed by CHOPCHOP (http://chopchop.cbu.uib.no/) and constructed using Transcript Aid T7 High Yield Transcription Kit (Thermo Fisher Scientific). Cas9 mRNA was generated by mMESSAGE mMACHINE T7 Transcription Kit (Invitrogen). Zebrafish carrying *mettl16* mutations were identified by Sanger sequencing. The stable *mettl16* mutant zebrafish line was obtained by several rounds of crossing and screening.

## Whole-mount in situ hybridization

Whole-mount in situ hybridization (WISH) was performed as described previously (Wei et al, 2014; Yu et al, 2019). All the templates of RNA probes were cloned from the cDNA library of wild-type embryo at 3 dpf and confirmed by Sanger sequencing. The digoxigenin-labeled RNA probes were synthesized using MAXIscript™ SP6 Transcription Kit (Invitrogen, United States). More than 48 zebrafish embryos were used to ensure that the proportion of homozygous embryos was above 12. WISH was conducted with three independent experiments. The images were captured using an optical microscope (BX53, Olympus). After imaging, DNA was extracted from the embryos for genotyping.

## qPCR and semi-qPCR

Zebrafish embryos at 3–5 dpf or cells were used for total RNA extraction. The heads were cut off and used for genotyping. For each of the samples, the above 15 tails (including CHT regions) were dissected and put together to extract RNA using the TRIzol Reagent (Vazyme) according to the manufacturer's protocol. RNA was reverse-transcribed into complementary DNA by TransScript All-in-One First-Strand cDNA Synthesis SuperMix. qRT-PCR was conducted with AceQ qPCR SYBR Green Master Mix

(Vazyme Biotech) in QuantStudio 3 or 6 pro-Flex Real-Time PCR system (Applied Biosystem). qPCR was performed with three biological replicates. For each sample, there were three technical replicates. Any technical replicate deviating more than 0.5cq (ct) from the other two replicates was excluded (Caballero-Solares et al, 2022; de Ronde et al, 2018). Gene expression was normalized to *actin* or *gapdh* and calculated using the $2^{-\Delta\Delta ct}$ method (Yuan et al, 2006):

average the Ct values for the technical replicates of each sample,

$$\Delta CT = \text{average CT(target)} - \text{average CT(reference gene)}$$

$$\triangle\triangle Ct = \triangle Ct\,(\text{Sample}) - \text{average}\,\triangle Ct\,(\text{Control})$$

$$\text{Fold gene expression} = 2^{-\triangle\triangle Ct}$$

For semi-qPCR, primers were designed for differentially spliced genes with aberrant splice sites. PCR products were separated by electrophoresis on a 1.5% agarose gel and photographed with XRS+ (Bio-Rad). The qPCR and semi-qPCR primers used are listed in Dataset EV5.

## Western blot

Zebrafish embryos at 3 dpf and 4 dpf or cells were collected for protein extraction. The heads of embryos were used for genotyping. About 20 tails of each genotype were collected together and lysed in RIPA lysis buffer with protease inhibitor. Western blot was performed as described previously(Yu et al, 2019; Zhang et al, 2014) with three biological replicates. The following primary antibodies were used in this study: anti-METTL16 antibody (Immunoway, Cat# YN3138), anti-GAPDH antibody (proteintech, Cat# 60004-1-1 g), anti-P53 antibody (GeneTex, Cat# GTX128135), anti-γH2AX antibody (Cell Signaling Technology, Cat# 2577), and anti-phospho-RB1 antibody (abclonal, Cat# AP0444). The images were captured by a ChemiDoc XRS1 gel imaging system (Bio-Rad). The gray value scanning of bands and calculation of protein abundance were done through Image J software.

## Immunofluorescence, EDU, and TUNEL staining

For immunostaining of whole-mount embryos, more than 48 embryos at 2 dpf, 3 dpf, 4 dpf, or 5 dpf were collected and fixed with 4% PFA overnight at 4 °C. After washing with PBST, embryos were digested with protease K for 20, 30, 40, and 50 min, respectively. Embryos were then blocked with PBST containing 10% goat serum at room temperature for 2 h. After blocking, embryos were incubated with anti-green fluorescent protein antibody (Dia-an, Cat# TAG0071) overnight at 4 °C, followed by goat-Alexa Fluor 488-conjugated anti-mouse secondary antibody (Abcam, 1:500) at 37 °C for 2 h. The heads were cut off and used for genotyping. The samples were imaged using a confocal microscope (FV3000, Olympus). Immunostaining was conducted with three independent experiments.

For EDU assay, more than 48 zebrafish embryos at 2 dpf, 3 dpf, 4 dpf, or 5 dpf were incubated in egg water containing 2 mM EDU (5-ethynyl-2-deoxyuridine) for 30 min at 4 °C. After rinsing three times in E3 medium, the embryos were transferred to freshwater raised for 2 h at 28.5 °C, and then fixed in 4% PFA at 4 °C overnight. After digestion with proteinase K (20 mg/mL) at room temperature for 20, 30, 40, and 50 min, respectively, cell proliferation was detected by the Cell-Light EDU Apollo567 In Vitro Kit (Ribobio, Guangzhou, China). The heads were cut off and used for genotyping. The samples were imaged using a confocal microscope (FV3000, Olympus). EDU assay was conducted with three independent experiments.

TUNEL staining from more than 48 zebrafish embryos at 2 dpf, 3 dpf, 4 dpf, or 5 dpf was performed using the TUNEL BrightRed Apoptosis Detection Kit (Vazyme, A113). The samples were imaged using a confocal microscope (FV3000, Olympus). TUNEL staining was conducted with three independent experiments.

## Morpholino (MO) and mRNA injections

The antisense morpholinos of *mettl16* were designed and purchased from GeneTools. The sequences are listed in Dataset EV5. The MO was diluted in RNase-free water, and then mixed with RNase-free phenol Red solution to achieve a concentration of 0.3 mM control MO or *mettl16* MO.

For rescue experiments, mRNAs were synthesized as previously described(Gu et al, 2019; He et al, 2015). The full-length coding sequences of zebrafish *mettl16* and *mybl2b* were amplified from cDNA samples of wild-type zebrafish embryos. The Mettl16 mutants with deleted nuclear localization signal and PP185/186AA mutants were constructed by Mut Express II Fast Mutagenesis Kit V2 (Vazyme C214-01). The cDNA fragments were subcloned into the pCS2+8CmCherry vector. Capped mRNAs were synthesized using the mMESSAGE mMACHINE SP6 Transcription Kit (Invitrogen, USA), and injected into the 1–2 cell stage embryos (100 pg).

## Cell lines

The human K562 (ATCC, CCL-243) and HEK293 (ATCC, CRL-1573) cell lines were cultured in RPMI 1640 medium (Gibco) and Dulbecco's modified Eagle's medium (DMEM, Gibco), respectively, both supplemented with 10% fetal bovine serum (FBS) and grown in 5% $CO_2$ at 37 °C. Penicillin-Streptomycin were supplemented in all media to prevent potential contamination.

## Primary cord blood sample

Human CD34$^+$ cells were isolated from the cord blood of healthy donors. Mononuclear cells (MNCs) were isolated by density gradient centrifugation with Ficoll (GE Healthcare Life Science). CD34$^+$ cells were enriched by positive selection using CD34 microbeads and a magnetic cell sorting system (Miltenyi).

Primary human cord blood CD34$^+$ cells were cultured as described before (Wang et al, 2020). Briefly, cells were cultured in StemSpan SFEM (StemCell Technologies) supplemented with rhIL3 (20 ng/mL), rhFLT3L (100 ng/mL), rhTPO (50 ng/mL), rhSCF (100 ng/mL), 1% penicillin/streptomycin (Sigma). All the cytokines were purchased from Peprotech.

## Cell sorting and flow analysis

Cell sorting was performed using Tg(*cmyb*:GFP) HSPC reporter zebrafish as previously described (Zhang et al, 2016b). Briefly, one-cell stage Tg(*cmyb*:GFP) embryos were injected with 0.3 mM control MO and *mettl16* MO and collected at 3 dpf, 4 dpf, and 5 dpf. The tails (including the CHT region) from approximately 200 embryos per group were dissociated, resuspended in 1 × DPBS with 5% heat-inactivated FBS, and sorted by MoFlo XDP.

To analyze cell cycle profiles, sorted GFP$^+$ cells from zebrafish and human cells were collected and washed twice with ice-cold PBS. After being fixed in 70% ethanol at −20 °C overnight, cells were resuspended in PBS, and stained with 1 mL propidium iodide solution (at a final concentration of 20 µg/mL propidium iodide, containing 0.1% Triton X-100 and 0.2 mg/mL RNase A) for 30 min protected from light. The flow cytometry data were obtained using CytoFlex (Beckman Coulter), and analyzed by FlowJo v10 software. Cell cycle analysis was conducted with three biological replicates.

## Lentivirus production and infection

To construct lentiviral vectors targeting *METTL16* or scrambled shRNA (NT), shRNAs were designed using the GPP web portal (Broad Institute). Primers are synthesized and cloned into pLKO.1 vector by digestion and ligation. The target sequence of sh*METTL16* is 5′-CGCAACAGAAGTGGATGATAT−3′(sh*METTL16*-1), 5′-CCAAAG TAACGTACACTGAAT−3′ (sh*METTL16*-2), and 5′-GCACCTACA TACGTAACCAAA-3′(sh*METTL16*-3); the non-targeting disruptive RNA sequence is 5′-CCTAAGGTTAAGTCGCCCTCG-3′(shNC). Lentiviral particles for delivery of lentiviral *METTL16* or NT vectors were produced using transfection with PEI. HEK293T cells were transfected with three plasmids: lentiviral METTL16 or NT constructs, psPAX2, and pVSV-G. 14 h post-infection, the medium was aspirated and replaced with 10 mL RPMI/10% FBS medium for subsequent treatment. Viral supernatants were collected at 36 and 60 h after transfection. Viral particles were transduced in K562 cells. Positive selection was carried out using puromycin-treated media (2 µg/mL).

## MeRIP-qPCR

For meRIP-qPCR, the procedure was modified from the previously reported methods (Chen et al, 2015; Weng et al, 2018). Total RNAs from the tails of zebrafish embryos at 3 dpf and human cells were extracted with TRIzol. For zebrafish samples, one-cell stage embryos were injected with 0.3 mM control MO and *mettl16* MO

and more than 200 zebrafish embryos were collected at 3 dpf. Anti-m⁶A polyclonal antibody (6 µg, Synaptic Systems, 202003) was incubated with 50 µL Protein A/G beads (Sigma, P9424) in 500 µL IPP buffer (150 mM NaCl, 0.1% NP-40, 10 mM Tris-HCl, pH 7.4, RNase Inhibitor) overnight at 4 °C. Then the total RNAs (20 µg) were incubated with the prepared antibody-beads mixture for 4 h at 4 °C. After washing with elution buffer (500 µL 0.1 M DTT, 0.0088 g NaCl, 50 µL PH 7.5 1 M Tris-HCl, 2 µL 0.5 M EDTA(PH8.0), 10 µL 10%SDS, 0.2 µL RNase Inhibitor) at 42 °C for 20 min, bound RNAs were extracted by TRIzol, and then reversely transcribed and amplified following the protocols described above. The enrichment of m⁶A was quantified using qPCR as reported(Zhang et al, 2017). The sequences of qPCR primers are listed in Dataset EV5.

For m⁶A level detection of the whole RNA, total RNAs from the CHT region of zebrafish embryos at 3 dpf were extracted with TRIzol. The heads were used for genotyping. For each of the samples, above 20 tails were collected. The RNAs were analyzed using the EpiQuik m⁶A RNA Methylation Quantification Kit (Colorimetric) (A&D Technology, A-P-9005).

## RIP-qPCR

RNA immunoprecipitation was performed as previously described with minor modifications (Chen et al, 2021; Shen et al, 2020). In brief, the tails of about 200 control MO and *mettl16* MO zebrafish at 3 dpf and human cells were placed in D-PBS with 5% heat-inactivated FBS and manually ground to prepare single-cell suspensions. Then, cells were lysed in RIP buffer (25 mM Tris pH 7.4, 150 mM KCl, 5 mM EDTA, 0.5% NP-40, 0.5 mM DTT, 100 U/mL RNasin and Protease Inhibitor Cocktail). About 10% of the lysates were reserved as input. For immunoprecipitation, cell lysates were incubated with 8 µg anti-METTL16 antibody (Abcam, Cat# ab252420), 8 µg anti-IGF2BP1 antibody (abclonal, Cat# A13581), or IgG antibody at 4 °C overnight, followed by addition of Protein A/G beads for immunoprecipitation at 4 °C for 2 h. After the beads were washed three times in RIP buffer, the co-precipitated RNAs were recovered from the beads with TRIzol. The detailed RNA extraction, cDNA synthesis, and qPCR processes were described above. qPCR was performed with three biological replicates. For each sample, there were three technical replicates. Briefly, fold enrichment was calculated over IgG adopting the 2^-ΔΔCT method, where ΔCt [normalized IP] = (Ct [IP] - (Ct [Input] - Log₂ (Input Dilution Factor))) and ΔΔCT = (ΔCt [normalized IP] – ΔCt [normalized IgG]). *thor* and *gapdh* were served as a positive and negative control for Igf2bp1 RIP in zebrafish (Hosono et al, 2017), respectively. *actin* was served as a negative control for Mettl16 RIP in zebrafish. *MALAT1* and *SRF* were served as positive control for METTL16 and IGF2BP1 RIP in human cells, respectively(Brown et al, 2016; Muller et al, 2019). The sequences of qPCR primers are listed in Dataset EV5.

## mRNA stability analysis

mRNA stability analysis was performed as previously described with minor modifications (Shima et al, 2017). About 20 control and *mettl16* morphant embryos were treated with 10 µg/mL actinomycin D (Sigma, A1410) or DMSO from 3 dpf for 4 and 8 h, respectively. Human cells were treated with 5 µg/mL actinomycin D

(Sigma, A1410) or DMSO for 4 and 8 h, respectively. The total RNAs were then extracted from the tails of zebrafish embryos and human cells for reverse transcription and qPCR. qPCR was performed with three biological replicates. For each sample, there were three technical replicates. At 4 and 8 h, the relative expression of *mybl2b* was calculated by normalizing the gene expression in actinomycin D-treated embryos or cells against that in DMSO-treated embryos or cells and calculated using the $2^{-\Delta\Delta ct}$ method (Zhang et al, 2017):

average the Ct values for the technical replicates of each sample,

$$\Delta CT = \text{average } CT(target) - \text{average } CT(reference\ gene)$$

$$\triangle\triangle Ct = \triangle Ct\,(actinomycin - D - treated) - \triangle Ct\,(DMSO - treated\ average)$$

Fold gene expression $= 2^{-\triangle\triangle Ct}$

## RNA-seq

Total RNAs from the tails (including CHT region) of siblings and *mettl16*⁻/⁻ embryos at 3 dpf were extracted using Trizol (Invitrogen) according to the manufacturer's protocol. The heads were used for genotyping. For each of the samples, above 20 tails were collected. RNA quality was assessed on an Agilent 2100 Bioanalyzer (Agilent Technologies,) and checked using RNase-free agarose gel electrophoresis. RNA-Seq was performed on an Illumina HiSeq2500 platform by Gene Denovo (Guangzhou, China). RNA-seq quality control and filtering were conducted using fastp software (version 0.18.0). Short read alignment tool Bowtie2 (version 2.2.8) was used for mapping reads to the ribosome RNA (rRNA) database. The rRNA-mapped reads were then removed. Paired-end clean reads were mapped to the zebrafish reference genome (Ensembl GRCz11) using HISAT2. 2.4. For each transcription region, an FPKM (fragment per kilobase of transcript per million mapped reads) value was calculated to quantify its expression abundance and variation using StringTie software. RNA differential expression analysis was performed by DESeq2 software using the following cut-off values: $P \leq 0.05$, |log2FC| ≥0.585. Alternative splicing was analyzed by rMATS (version 4.0.1) and indicated as PSI (percent-spliced-in) values using the following cut-off values: $P < 0.05$, |delta PSI| ≥0.2. Hiplot software (https://hiplot-academic.com/) was used to identify Gene Set Enrichment Analysis (GSEA) terms enriched in the differentially expressed genes.

## MeRIP-seq

Total RNAs from the tails (including CHT region) of siblings and *mettl16*⁻/⁻ embryos at 3 dpf were isolated using TRIzol reagent (Invitrogen). The heads were used for genotyping. For each of the samples, above 200 tails were collected. The RNA quality and quantity of each sample were assessed using NanoDrop ND-1000 and Bioanalyzer 2100 (Agilent). Total RNAs (20 µg) were depleted of ribosomal RNA by rRNA Depletion Kits (Thermo Fisher Scientific), and Magnesium RNA Fragmentation Module (NEB, cat.e6150) was used to randomly fragment RNA. Anti-m⁶A antibody (202003, Synaptic Systems) was applied for m⁶A immunoprecipitation as described above. Then, the RNA was reverse-transcribed to create the cDNA by SuperScript™ II Reverse Transcriptase (Invitrogen, cat.1896649), and the Illumina Novaseq™ 6000 (LC-Bio Technology Co., Ltd.) was used to perform the paired-end sequencing. The sequence quality of IP and input

samples was verified using fastp software. HISAT2 was used to map reads to the zebrafish reference genome (Ensembl GRCz11). Mapped reads of IP and input libraries were provided for the R package exomePeak, which identifies m⁶A peaks with bed or bigwig format that can be adapted for visualization on the IGV software (http://www.igv.org). A stringent cutoff threshold for $P \leq 0.05$, |log2FC|≥1 was used to obtain high-confidence peaks. MEME and HOMER were used for de novo and known motif finding followed by localization of the motif with respect to peak summit. Called peaks were annotated by intersection with gene architecture using the R package ChIPseeker.

## Statistical analysis

Data were analyzed with GraphPad Prism 10 and were presented as mean ± SEM as indicated. Statistical significance was calculated using two-tailed unpaired Student's *t*-tests, one-way ANOVA, or two-way ANOVA. Adjusted *P* values <0.05 (*P* values generated from *t*-tests were corrected using the Benjamini–Hochberg method) were considered statistically significant unless otherwise specified. All experiments were performed with at least three independent biological or experimental replicates. For every figure, the statistical tests are justified as appropriate. Detailed information about the statistical methods used is specified in the figure legends or Methods.

# Data availability

The m⁶A-seq and RNA-seq data have been deposited in the gene expression omnibus (GEO) with the accession number GSE225137. All data were available to the authors upon request.

# Peer review information

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

## Acknowledgements

We express our gratitude to Prof. Feng Liu (IOZ, CAS, China) and Dr. Luying Liu (Yale Uni, USA) for their critical reading and valuable comments. We also appreciate the valuable advice from our colleagues, Drs. Danna Jia, Yuexia Lv, and Jiayi Tu. This work was supported by grants from the Ministry of Science and Technology of China (No.2018YFA0801000), the National Natural Science Foundation of China (32270646 to M.L., 82071010 to Z.T., 31922085 to D.L., 81870959 to X.Z., and 32370880 to S.Y.) and Program for HUST Academic Frontier Youth Team (2016QYTD02 to X.Z.).

## Author contributions

**Yunqiao Han**: Conceptualization; Data curation; Software; Formal analysis; Validation; Investigation; Visualization; Methodology; Writing—original draft. **Kui Sun**: Validation; Investigation; Visualization; Methodology. **Shanshan Yu**: Conceptualization; Resources; Funding acquisition; Methodology; Writing—review and editing. **Yayun Qin**: Resources; Formal analysis; Investigation; Methodology. **Zuxiao Zhang**: Validation. **Jiong Luo**: Software. **Hualei Hu**: Software. **Liyan Dai**: Validation. **Manman Cui**: Methodology. **Chaolin Jiang**: Methodology. **Fei Liu**: Methodology. **Yuwen Huang**: Investigation. **Pan Gao**: Methodology. **Xiang Chen**: Formal analysis. **Tianqing Xin**: Validation. **Xiang Ren**: Writing—review and editing. **Xiaoyan Wu**: Resources. **Jieping Song**: Resources. **Qing Wang**: Writing—review and editing. **Zhaohui Tang**: Funding acquisition; Writing—review and editing. **Jianjun Chen**: Writing—review and editing. **Haojian Zhang**: Resources; Methodology; Writing—review and editing. **Xianqin Zhang**: Conceptualization; Funding acquisition; Methodology; Writing—review and editing. **Mugen Liu**: Conceptualization; Data curation; Supervision; Funding acquisition; Methodology; Project administration; Writing—review and editing. **Daji Luo**: Conceptualization; Resources; Data curation; Supervision; Funding acquisition; Methodology; Project administration; Writing—review and editing.

## Disclosure and competing interests statement

The authors declare no competing interests.

# Expanded View Figures

**Figure EV1.  Mettl16 governs HSPC maintenance in the CHT in a direct and cell-autonomous fashion.**

(**A**) Expression of the HSPC markers *runx-1* and *cmyb* in the hemogenic endothelium of the ventral dorsal aorta in *mettl16* mutants by WISH at 28 hpf, 36 hpf, and 60 hpf, respectively. Numbers at the bottom right indicate the number of embryos with similar staining patterns among all embryos examined. $n = 3$ independent experiments. Scale bars, 100 μm. (**B**) Quantification of live imaging in Fig. 2J. '*n*' indicates the number of individuals analyzed for GFP$^+$ cells in the CHT of siblings and *mettl16* mutants at 2 and 5 dpf, respectively. (**C**) WISH analysis showing the expression of arterial markers *dll4, efnb2a* and venous markers *dab2* in *mettl16* mutants at 30 hpf. Numbers at the bottom right indicate the number of embryos with similar staining patterns among all embryos examined. $n = 3$ independent experiments. Scale bars, 50 μm. (**D**) WISH analysis showing the expression of vascular markers *flk1* in *mettl16* mutants at 3 dpf (upper). Scale bars, 100 μm. Live imaging of artery and vein in the aorta-gonad-mesonephros (AGM) and CHT region within Tg (*flk1*: EGFP) background in siblings and *mettl16$^{-/-}$* embryos at 30 hpf-4 dpf (lower). A artery, V vein. Scale bars, 40 um. Numbers at the bottom right indicate the number of embryos with similar staining patterns among all embryos examined. $n = 3$ independent experiments. (**E**) Dot plot showing the expression of *mettl16* in *kdrl$^+$* cluster in Fig. 1E. (**F**) The spatiotemporal expression pattern of *mettl16* in the CHT analyzed from GSE120581. HSC hematopoietic stem cell, EC endothelial cell, HE hemogenic endothelium, NC non-endothelial and nonhematopoietic cells. (**G**) The expression of *desma* (a somite marker), *ifabp* (an intestine marker) and *huc* (a neuro marker) in sibling and *mettl16$^{-/-}$* mutant embryos at 4 dpf. Numbers at the bottom right indicate the number of embryos with similar staining patterns among all embryos examined. $n = 3$ independent experiments. Scale bars, 100 μm. Data information: In (**B**), data were represented as mean ± SEM, ****adjusted $P < 0.0001$, n.s. non-significant, Student's *t*-test.

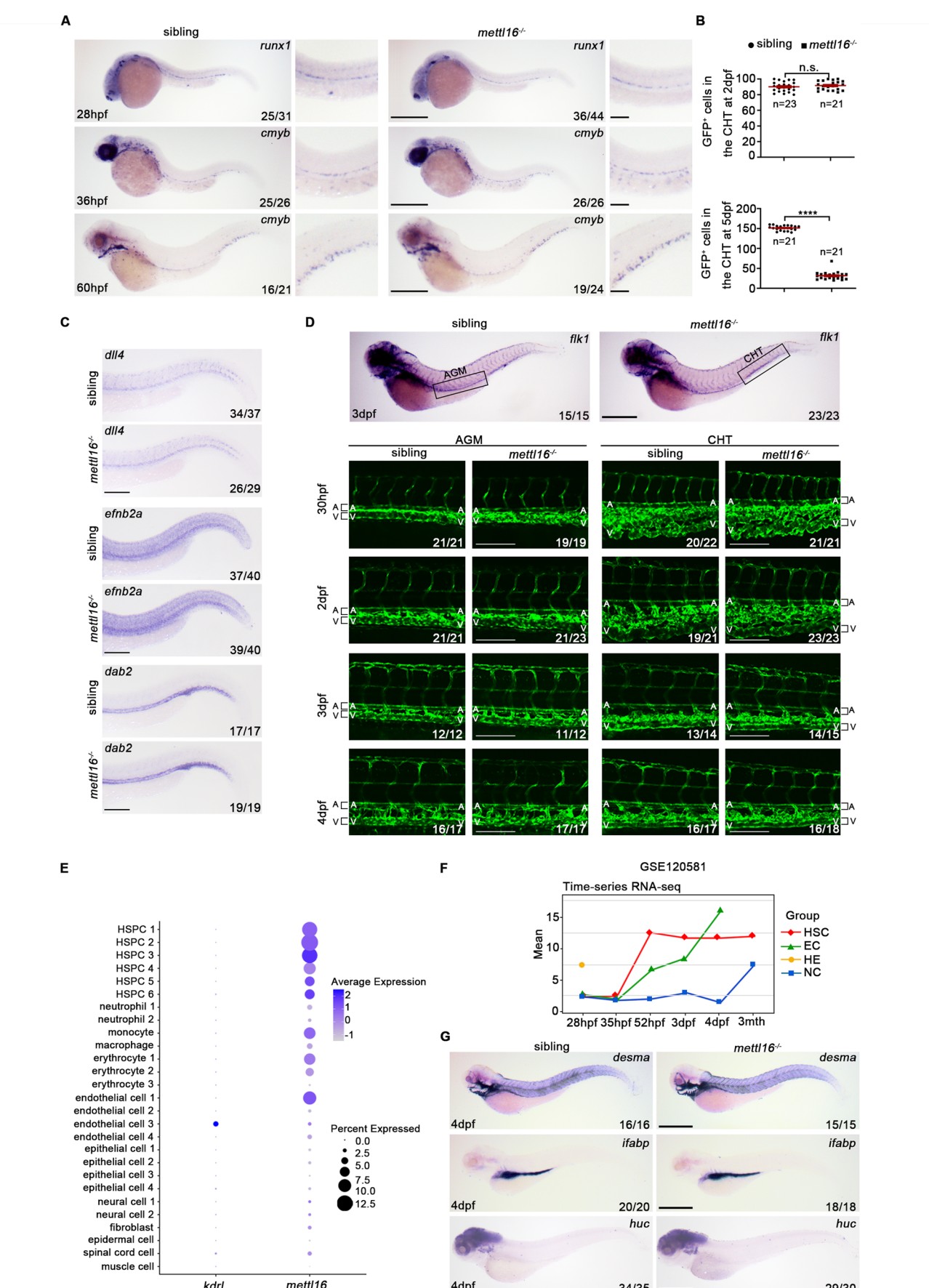

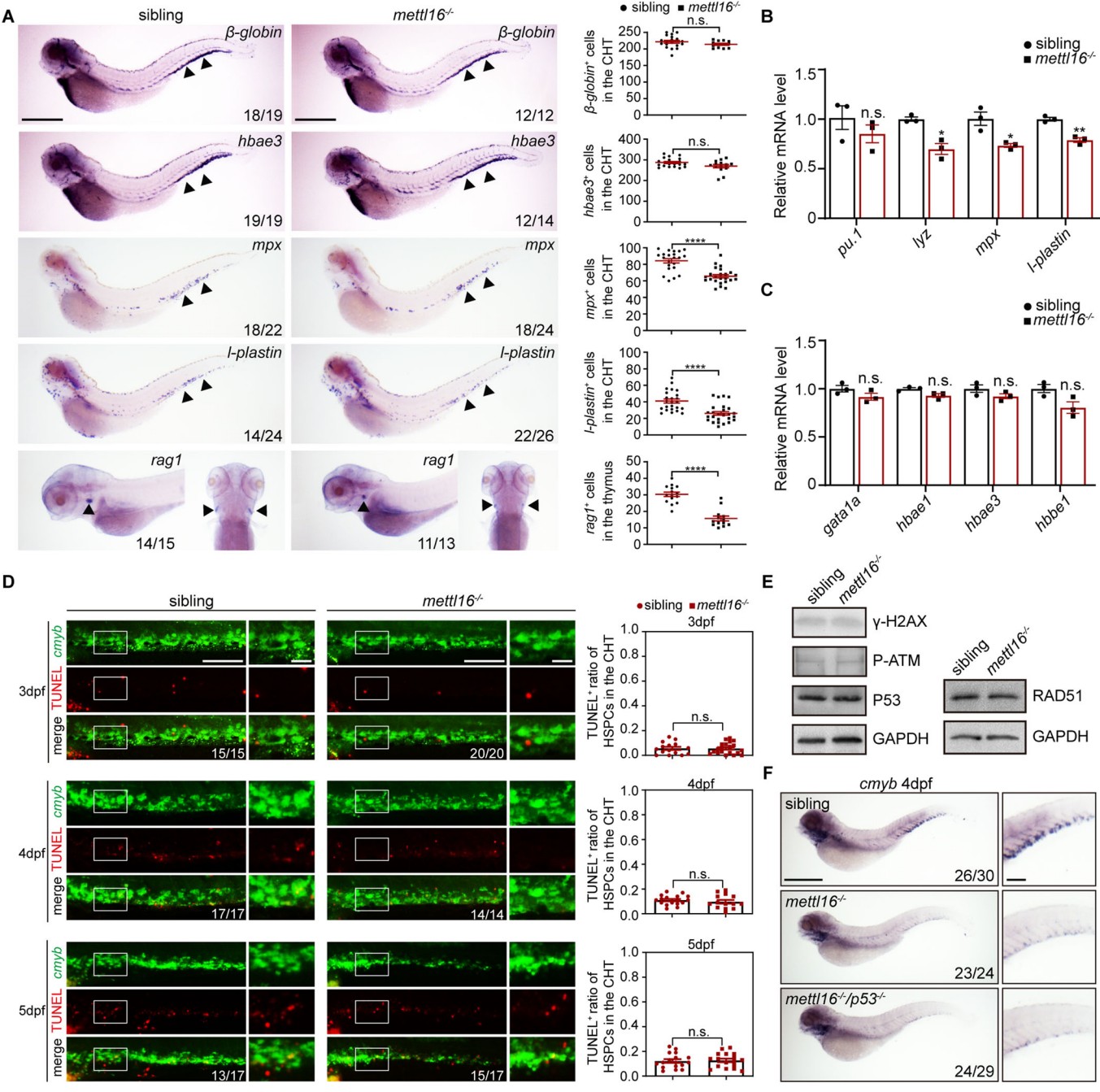

**Figure EV2. Loss of Mettl16 does not cause excessive differentiation or apoptosis of HSPCs.**

(A) The expression of erythroid cell markers *b-globin* and *hbae3*, myeloid cell markers *mpx* and *l-plastin* at 3 dpf, as well as lymphocytes marker *rag1* at 4 dpf in siblings and *mettl16* mutants by WISH. Numbers at the bottom right indicate the number of embryos with similar staining patterns among all embryos examined. $n = 3$ independent experiments. The black arrowheads indicate differentiated blood cells. Scale bars, 100 μm. (B, C) qPCR showing the expression of myeloid cell markers (B) and erythroid cell markers (C) in siblings and *mettl16* mutants at 3 dpf. $n \geq 15$ per group, performed with three biological replicates. (D) Double immunostaining of *cmyb*: EGFP and TUNEL showing the number of apoptotic HSPCs in siblings and *mettl16* mutants from 3 to 5 dpf. Numbers at the bottom right indicate the number of embryos with similar staining patterns among all embryos examined. $n = 3$ independent experiments. Scale bars, 40 μm. (E) Western blot showing the protein expression level of γ-H2AX, P-ATM, P53, and RAD51 in siblings and *mettl16* mutants at 4 dpf. $n \geq 20$ per group, performed with three biological replicates. (F) Effect of restoration of P53 deletion on the expression of HSPC marker *cmyb* in *mettl16*$^{-/-}$ embryos at 4 dpf. Numbers at the bottom right indicate the number of embryos with similar staining patterns among all embryos examined. $n = 3$ independent experiments. Scale bars, 100 μm. Data information: In (A–D), data were represented as mean ± SEM, *adjusted $P < 0.05$, **adjusted $P < 0.01$, **** adjusted $P < 0.0001$, n.s. non-significant, Student's *t*-test.

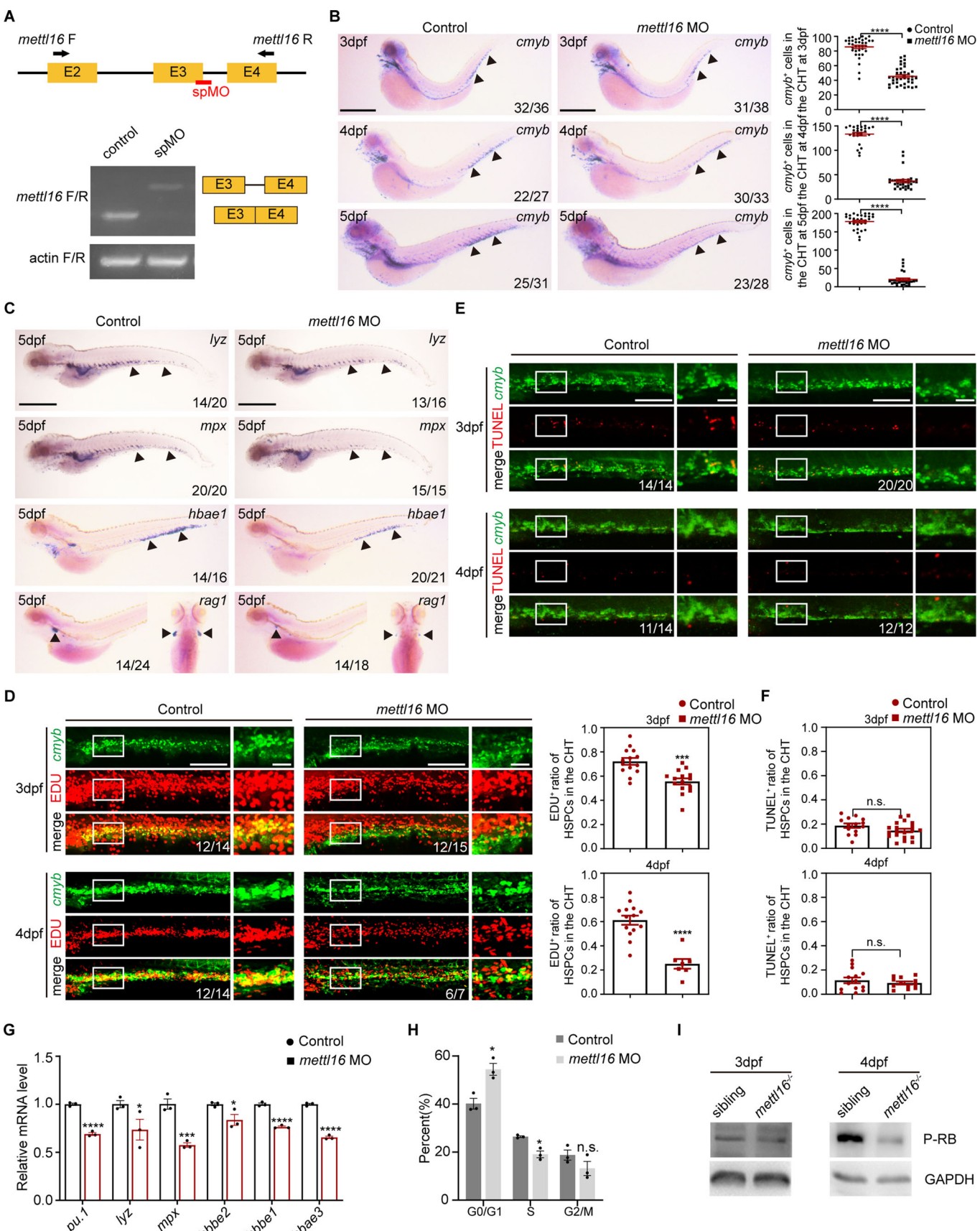

◀ **Figure EV3.** *mettl16* **morpholino reproduces the** *mettl16* **knockout effect on HSPCs.**

(A) The *mettl16* splice MO was designed to target the boundary of exon3 and intron 3 (red line marks the target region). The primers were used to amplify the exon region for PCR validation of this MO. (B) WISH assay showing the number of HSPCs in the CHT of *mettl16* morphants from 3 dpf to 5 dpf. Numbers at the bottom right indicate the number of embryos with similar staining patterns among all embryos examined. $n = 3$ independent experiments. The black arrowheads indicate *cmyb*$^+$ cells in the CHT. Scale bars, 100 µm. (C) Expression of definitive hematopoiesis markers-*lyz* (myeloid), *mpx* (myeloid), *hbae1* (erythroid), and *rag1* (lymphoid) in *mettl16* morphants compared with control at 5 dpf. Numbers at the bottom right indicate the number of embryos with similar staining patterns among all embryos examined. $n = 3$ independent experiments. The black arrowheads indicate differentiated blood cells. Scale bars, 100 µm. (D) Double immunostaining of *cmyb*: EGFP and EDU showing the proliferation of HSPCs in the CHT of morphants at 3 dpf and 4 dpf. Numbers at the bottom right indicate the number of embryos with similar staining patterns among all embryos examined. $n = 3$ independent experiments. Scale bars, 40 µm. (E, F) Double immunostaining of *cmyb*: EGFP and TUNEL showing the number of apoptotic HSPCs at 3-4 dpf. Numbers at the bottom right indicate the number of embryos with similar staining patterns among all embryos examined. $n = 3$ independent experiments. Scale bars, 40 µm. (G) qPCR showing the expression of myeloid cell markers (*pu.1*, *lyz*, and *mpx*) and erythroid cell markers (*hbbe2*, *hbbe1*, and *hbae3*) in *mettl16* morphants compared with control at 4 dpf. $n \geq 15$ per group, performed with three biological replicates. (H) Flow analysis showing the cell cycle of HSPCs of *mettl16*-deficient zebrafish at 5 dpf. $n \geq 200$ per group, performed with three biological replicates. (I) The protein expression level of P-RB in *mettl16*$^{-/-}$ embryos compared with siblings at 3-4 dpf. $n \geq 20$ per group, performed with three biological replicates. Data information: In (B, D, F, G, H), data were represented as mean ± SEM, *adjusted $P < 0.05$, ***adjusted $P < 0.001$, ****adjusted $P < 0.0001$, n.s. non-significant, Student's *t*-test.

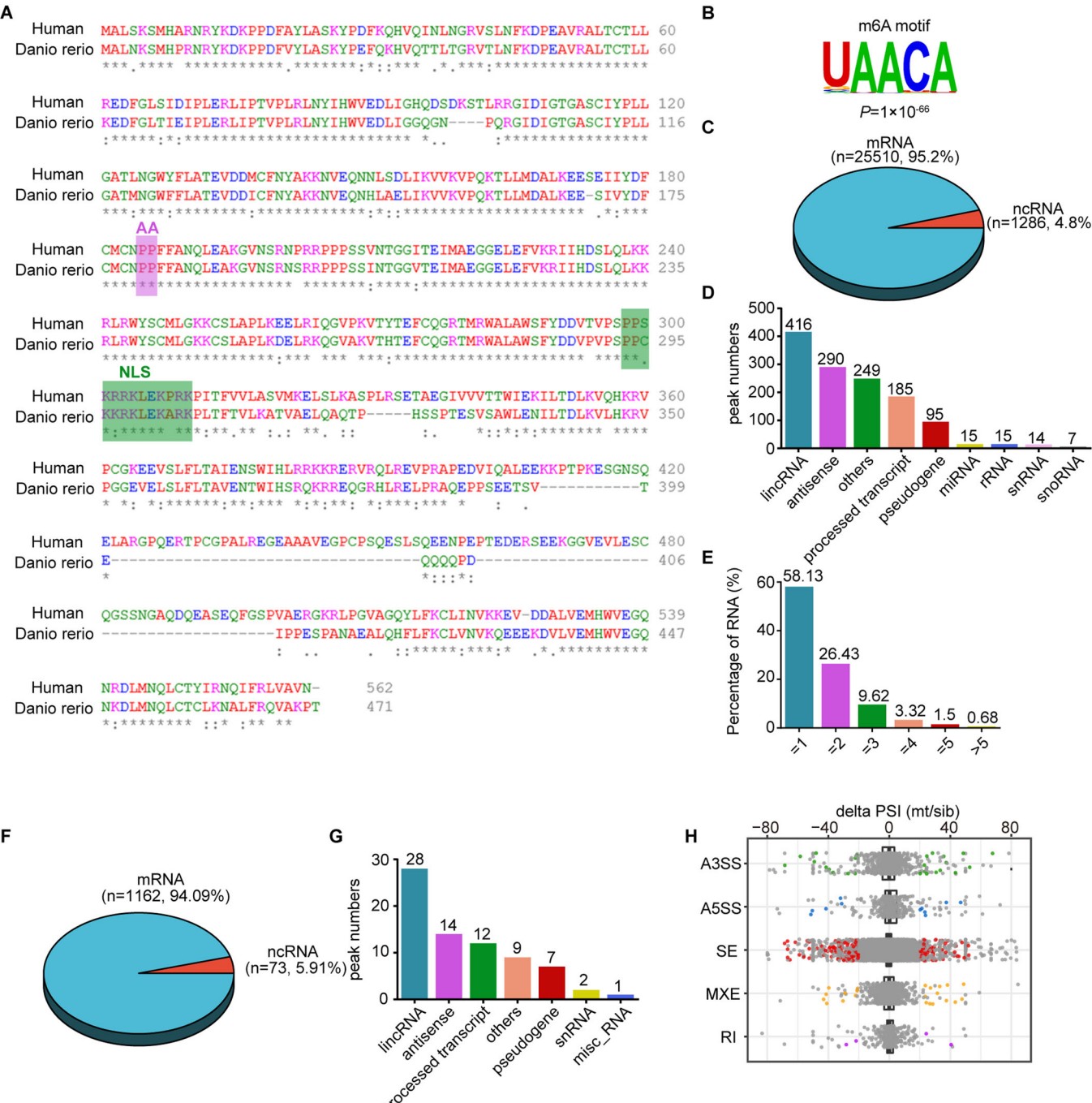

**Figure EV4. Disruption of Mettl16 invokes the alteration of m⁶A methylome and transcriptome.**

(A) Protein sequence alignment of METTL16 proteins between zebrafish and humans. The pink rectangle indicates the site of PP180/181AA, and the green rectangle indicates the site of predicted nuclear localization signal (NLS). (B) Sequence motif identified within m⁶A peaks by HOMER database in siblings. (C) Percentage of mRNAs and non-coding RNAs containing m⁶A peaks in siblings. (D) Bar plots showing the number of representative non-coding RNAs containing m⁶A peaks in siblings. (E) Percentage of m⁶A-methylated mRNAs with different numbers of m⁶A peaks in siblings. (F) Percentage of mRNAs and non-coding RNAs containing hypomethylated m⁶A peaks in *mettl16* mutants. (G) Bar plots showing the number of representative non-coding RNAs containing hypomethylated m⁶A peaks in *mettl16* mutants. (H) Categories of differentially spliced genes based on the changed PSI value in the *mettl16* mutants (mt). The percent-spliced-in index (PSI) indicates the efficiency of splicing a specific exon into the transcript population of a gene. A3SS alternative 3′ splice site, A5SS alternative 5′ splice site, SE skipped exon, RI retained intron, MXE mutually exclusive exon.

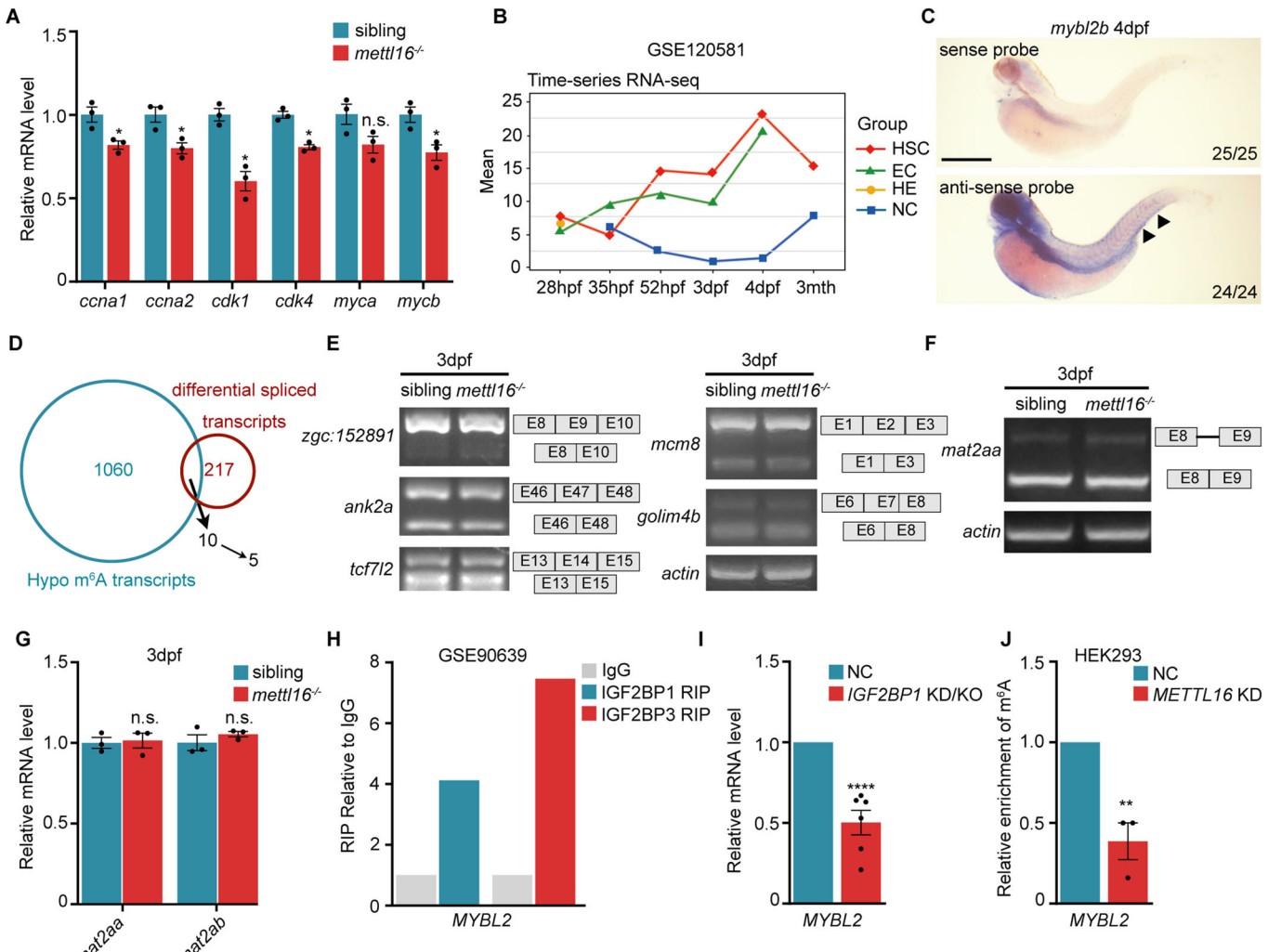

**Figure EV5. METTL16 regulates the expression of *MYBL2* in an m⁶A-dependent manner mediated by IGF2BP1.**

(A) qPCR analysis showing the mRNA expression level of the downstream targets associated with G1/S arrest of *mybl2b* in *mettl16⁻/⁻* embryos at 3 dpf. $n \geq 15$ per group, performed with three biological replicates. (B) The spatiotemporal expression pattern of *mybl2b* in the CHT generated from GSE120581. HSC hematopoietic stem cell, EC endothelial cell, HE hemogenic endothelium, NC non-endothelial and nonhematopoietic cells. (C) WISH analysis of *mybl2b* spatiotemporal expression in siblings at 4 dpf. Numbers at the bottom right indicate the number of embryos with similar staining patterns among all embryos examined. $n = 3$ independent experiments. The black arrowheads indicate *mybl2b⁺* cells in the CHT. Scale bars, 100 μm. (D) Venn diagram showing the overlap between differentially spliced genes ($P < 0.05$, |delta PSI| ≥0.2) and hypomethylated transcripts in *mettl16* mutants compared with siblings. Statistical analysis was performed by likelihood-ratio test. (E) Semi-qPCR analysis showing the splicing of G0/G1 arrest-related genes with differential splicing and hypomethylated m⁶A peaks in *mettl16* mutants. $n \geq 15$ per group, performed with three biological replicates. (F) Semi-qPCR analysis showing the splicing of *mat2aa* in *mettl16* mutants at 3 dpf. $n \geq 15$ per group, performed with three biological replicates. (G) qPCR analysis showing the mature mRNA expression level of *mat2aa* and *mat2ab* in *mettl16* mutants at 3 dpf. $n \geq 15$ per group, performed with three biological replicates. (H) IGF2BP1 and IGF2BP3 can bind *MYBL2* mRNA directly in HEK293 cells by RIP-seq (GSE90639). (I) RNA-seq analysis showed the mRNA expression level of *MYBL2* in *IGF2BP1* knockdown or knockout cells. Each data point in the figure corresponds to independent data sourced from GSE146546, GSE146803, GSE161087, GSE158258, GSE133097, and GSE161086, respectively. (J) m⁶A-seq analysis showed the m⁶A level of *MYBL2* upon *METTL16* knockdown in HEK293 cells. Each data point in the figure corresponds to three biological replicates from GSE90914. Data information: In (A, G, I, J), data were represented as mean ± SEM, *adjusted $P < 0.05$, **adjusted $P < 0.01$, **** adjusted $P < 0.0001$, n.s. non-significant, Student's *t*-test.

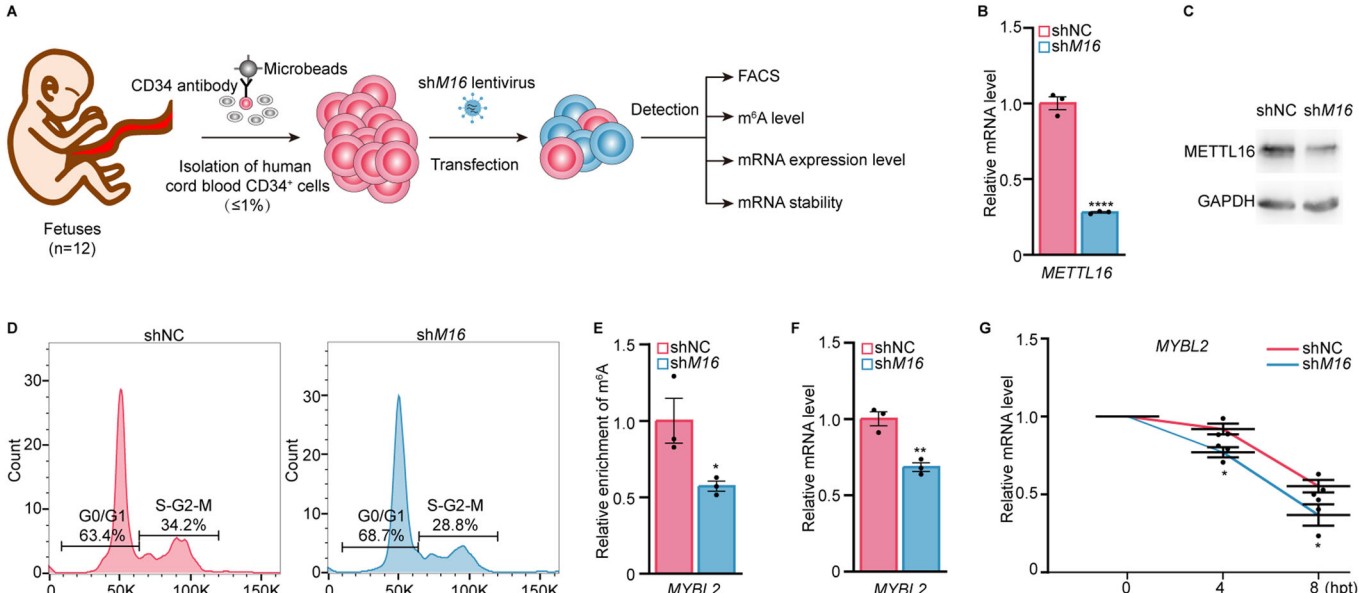

**Figure EV6. The METTL16/m⁶A/*MYBL2* axis in G1/S progression is conserved in human cord blood CD34⁺ cells.**

(A) Flow chart of human cord blood CD34⁺ cell isolation and lentivirus transduction. (B) Validation of the knockdown (KD) efficiency of the shRNA against *METTL16* by qPCR in CD34⁺ cord blood cells derived from healthy donors. $n = 3$ biological replicates, 2 healthy donors per group. (C) Validation of the KD efficiency of the shRNA against *METTL16* by western blot in CD34⁺ cord blood cells derived from healthy donors. $n = 2$ biological replicates, 2 healthy donors per group. (D) Flow analysis showing increased G0/G1 phase in CD34⁺ cord blood cells derived from healthy donors after *METTL16* knockdown. $n = 2$ biological replicates, 2 healthy donors per group. (E) m⁶A enrichment in *MYBL2* mRNA in CD34⁺ cord blood cells derived from healthy donors after *METTL16* knockdown by meRIP-qPCR. $n = 3$ biological replicates, 2 healthy donors per group. (F) qPCR analysis showing that the mRNA expression level of *MYBL2* in CD34⁺ cord blood cells derived from healthy donors after *METTL16* knockdown. $n = 3$ biological replicates, 2 healthy donors per group. (G) qPCR analysis of CD34⁺ cord blood cells derived from healthy donors treated with actinomycin D for 4 and 8 h showing accelerated *MYBL2* mRNA degradation after *METTL16* knockdown. $n = 3$ biological replicates, 2 healthy donors per group. Data information: In (B, E–G), data were represented as mean ± SEM, *adjusted $P < 0.05$, **adjusted $P < 0.01$, **** adjusted $P < 0.0001$, Student's *t*-test (B, E, F), two-way ANOVA analysis with post hoc test of Tukey's multiple comparison correction (G).

