## [Peer Review File · The EMBO Journal]

A Mettl16/m6A/Mybl2b/Igf2bp1 axis ensures cell cycle progression of embryonic hematopoietic stem and progenitor cells

Mugen Liu, Yunqiao Han, Kui Sun, Shanshan Yu, Yayun Qin, Zuxiao Zhang, Jiong Luo, Hualei Hu, Liyan Dai, Manman Cui, Chaolin Jiang, Fei Liu, Yuwen Huang, Pan Gao, Xiang Chen, Tianqing Xin, Xiang Ren, Xiaoyan Wu, Jieping Song, Qing Wang, Zhaohui Tang, Jianjun Chen, Haojian Zhang, Xianqin Zhang, and Daji Luo

Corresponding authors: Mugen Liu (lium@hust.edu.cn) , Xianqin Zhang (xqzhang04@hust.edu.cn), Daji Luo (luodaji@ihb.ac.cn)

Review Timeline:

Submission Date:	17th Jul 23
Editorial Decision:	21st Aug 23
Revision Received:	11th Jan 24
Editorial Decision:	1st Mar 24
Revision Received:	6th Mar 24
Accepted:	7th Mar 24

Editor: Kelly Anderson

Transaction Report:

Dear Prof. Liu,

Thank you for submitting your manuscript for consideration by the EMBO Journal.

As you will see from the enclosed reports, there are considerable concerns regarding whether the mechanism proposed is supported by the current data, which we feel will require longer than standard revision time to address.

Given these opinions and the fact that the EMBO Journal can only afford to accept papers which receive enthusiastic support from a majority of referees, I'm afraid we cannot offer to publish it. However, in this case both referees do acknowledge the potential of this work. Therefore, should you be able fully address the concerns raised by both referees, we would be open to reconsidering a revised version. In that case, we would also reassess the novelty of the work.

I am sorry that I can't be more positive on this occasion for the EMBO Journal, but I hope you nonetheless find the comments useful.

Yours sincerely,

Kelly M Anderson, PhD
Editor, The EMBO Journal
k.anderson@embojournal.org

Referee #1:

In this manuscript the authors show that *Mettl16* is highly expressed in embryonic HSPCs in many vertebrates. Using zebrafish, they showed that *mettl16* is not required for the specification of HSPCs *in vivo* but for their expansion and maintenance in the CHT in a cell intrinsic manner. This is due to defective proliferation without any increase in differentiation or apoptosis. This was also tested in a morphant. The cell cycle arrest was not due to DNA damage or p53 signaling. Mutation analysis showed that the defective HSPC proliferation was due to the methyltransferase activity of *Mettl16*. Indeed, m6A levels were notably decreased in *mettl16* deficient embryos. MeRIP and RNAseq results showed many transcripts that had less m6A methylation or deregulated expression respectively. A direct target of *Mettl16*, *mybl2b* was found responsible for the phenotype. *Igf2bp1* is a potential m6A reader of *mybl2b*, mediating the effect of *Mettl16* on *mybl2b* mRNA stability via m6A modification.

Overall this is a nice and complete manuscript. Some minor suggestions:

- Can the authors check arterial development and not only vessel development?
- Can the authors put number of embryos in all the *in situ* and fluorescent photos
- Explain this statement: "The proliferation signals of HSPCs in the CHT of *mettl16*^{-/-} mutants were comparable with siblings at 2dpf (Fig 3A)." What signals?
- Compare MeRIP and RNA seq results and discuss further the results of this comparison
- Can the authors show the conservation of the mechanism in a non leukemic system like in HPC7 cells?
- Figure with MeRIP-seq tracks. Please show also the gene and not only write the exon.
- The difference in the cycling cells is small (approximately 3% difference in S phase cells). Can the authors comment on how this difference can lead to such severe phenotype?
- Why the effect of *Mettl16* is cell intrinsic to HSPCs since it seems such a generic phenotype. Can the author use another cell type (non hematopoietic) to check whether this is happening also in non-blood cells?
- Is *mettl16* expressed in *kdrl*⁺ cells?

Referee #2:

The authors report that the production of HSPCs is regulated by *METTL16*, and that *mettl16*^{-/-} zebrafish embryos fail to develop normal numbers of HSPCs. The authors analyzed the molecular consequences of the lack of *METTL16* and concluded that *mettl16* deletion resulted in G1/S progression block. Molecular analysis of the potential *METLL16* targets identified potential molecular targets (e.g., *mybl2b*), and suggested a mechanism (half-life regulation), which the authors indicated to be conserved in mammalian cell lines.

The study is important, and the phenotype that is reported in figure 1 is convincing. However, the data obtained from molecular analysis is insufficient for drawing the conclusions that are made in the paper. My biggest concern is that many of the reported significantly different results are actually insignificant. Unless I am misreading the supplementary datasets, some results are simply insignificant, but treated in the text as significant (e.g., mybl2b expression). Other assays (e.g., qPCRs, WB, see below), report statistically significant results, but with unreasonable low effect sizes (e.g., 10-20%; see below). In summary, despite the interesting phenotype, the analysis is not well-supported by the data.

Major comments

Clarification of the qPCR analyses throughout the paper is required, and preferably, presentation of the source data. Key figures showing gene expression analyses by qPCR appear overoptimistic. For example, detection of significant ~10-15% differences in expression (e.g., Fig EV2C). Detection of such differences between a control and condition in qPCR is a nearly impossible task, requiring controls that are unavailable. Lack of description of the number of replicates used (biological and technical) does not add confidence. Lack of variance in the expression of the controls is unexpected, and suggests that the analysis was done incorrectly. Correction for multiple hypotheses testing is required and could impact some of the assertions made.

The lack of description of the number of replicates used in various assays, except for a brief indication in line 515, makes some of the data challenging to assess. Much improved description of experimental data collection and analysis is required, specifically for qPCR, WB, EDU, TUNEL, FACS, RIP-qPCR, mRNA stability analysis.

The claim that there is a G1/S block in the *mettl16*^{-/-} is not well supported. The FACS analysis (3 dpf) shows a mild difference in the cell cycle state of cells that could not account for the very big difference in *cmyb*⁺ cells that appears between 2 dpf and 3 dpf. The EDU and pH3⁺ HSPCs labeling is convincing, but is G1/S arrest the most likely interpretation? Supporting data shown in Fig 3E is insufficient: First, it is debatable that western blots can detect <20% difference in protein abundance and with such a remarkably low variability. Second, even if the difference in protein abundance is real, a valid interpretation is that it is an outcome of cells being at different cell states (which does not require them to be in a cell cycle arrest).

The analysis of *mybl2b* gene expression is inconsistent with the data. Dataset EV3 (gene expression) indicates that *mybl2b* is not significantly differentially expressed in *mettl16*^{-/-} (FDR=0.74). The effect size in the qPCR analysis is similar to the RNAseq (~35%). The qPCR shows a significant result, but has the same issues indicated above (lack of variance in controls, no multiple hypotheses correction). Similarly, the decrease in expression of *mybl2b* targets (Fig 5C) is unsupported by the RNAseq data (dataset EV3). The RIP-qPCR experiments lack essential controls for determining the specificity of the assay. The conclusion (line 263-265): "these results demonstrate that *mybl2b* mRNA is a bona fide direct m6A target of *Mettl16*, and *Mettl16* regulates mRNA stability of *mybl2b* through m6A reader protein *Igf2bp1*" is too strong. It is possible, but the data that supports this assertion is weak.

Minor comments;

The distribution *cmyb*⁺ cell number in the wildtype (control in Fig 4C and 4D) is different from the distribution shown in figure 1 for the same time point. Why are these different?

The RIP-qPCR experiments shown in figure 6B,D require better description of the methods, and controls (e.g., controlling for enrichment by using negative and positive controls).

The authors need to better compare and discuss their results with the analysis of HSPC development in *mettl3* deficient zebrafish embryos.

Line 64: is it ablation of *mettl14*? Or *mettl14* directly inhibits differentiation of HSPCs.

Line 100: missing acronym

** As a service to authors, EMBO Press provides authors with the possibility to transfer a manuscript that one journal cannot offer to publish to another EMBO publication or the open access journal Life Science Alliance launched in partnership between EMBO Press, Rockefeller University Press and Cold Spring Harbor Laboratory Press. The full manuscript and if applicable, reviewers' reports, are automatically sent to the receiving journal to allow for fast handling and a prompt decision on your manuscript. For more details of this service, and to transfer your manuscript please click on Link Not Available. **

Point-by-point responses:**Referee #1:**

In this manuscript, the authors show that *Mettl16* is highly expressed in embryonic HSPCs in many vertebrates. Using zebrafish, they showed that *mettl16* is not required for the specification of HSPCs *in vivo* but for their expansion and maintenance in the CHT in a cell intrinsic manner. This is due to defective proliferation without any increase in differentiation or apoptosis. This was also tested in a morphant. The cell cycle arrest was not due to DNA damage or p53 signaling. Mutation analysis showed that the defective HSPC proliferation was due to the methyltransferase activity of *Mettl16*. Indeed, m⁶A levels were notably decreased in *mettl16* deficient embryos. MeRIP and RNAseq results showed many transcripts that had less m⁶A methylation or deregulated expression respectively. A direct target of *Mettl16*, *mybl2b* was found responsible for the phenotype. *Igf2bp1* is a potential m⁶A reader of *mybl2b*, mediating the effect of *Mettl16* on *mybl2b* mRNA stability via m⁶A modification.

Overall this is a nice and complete manuscript. Some minor suggestions:

Response: We sincerely appreciate your positive and constructive feedback on the manuscript. Your valuable suggestions have significantly enhanced the quality of this work. Thank you very much.

1.Can the authors check arterial development and not only vessel development?

Response: We examined the arterial development using arterial marker *dll4* and *efnb2a* by whole-mount in situ hybridization (WISH) at 30hpf, when HSPCs emerge. The arterial development was normal in *mettl16*^{-/-} zebrafish (New Figure EV1C). Unfortunately, the probes of arterial markers were hard to stain the embryos at the later developmental stage by WISH. Hence, we performed a time course analysis of arterial development from 30hpf to 4dpf using Tg (*flk1*: EGFP) zebrafish instead. Confocal live imaging showed that the arterial development was unchanged in both aorta-gonad-mesonephros (AGM) and caudal hematopoietic tissue (CHT) regions of *mettl16*^{-/-} zebrafish until 4dpf (New Figure EV1D). I hope that you are satisfied with our revision.

New Figure EV1C and D

2.Can the authors put number of embryos in all the in situ and fluorescent photos

Response: In this revision, we have put the number of embryos in all the in situ and fluorescent photos (Fig 2A-H, 2J, 2L, 3A, 3B, 4C, 5H; Fig EV1A, 1C, 1D, 1G, 2A, 2D, 2F, 3B-E, 5C and Appendix Fig S2).

3.Explain this statement: "The proliferation signals of HSPCs in the CHT of *mettl16^{-/-}* mutants were comparable with siblings at 2dpf (Fig 3A)." What signals?

Response: We are sorry for the unclear statement in this sentence. The signals indicate the proliferating HSPCs, as depicted by the yellow dots (with white arrowheads) in Figure 3A. We have made revision to enhance clarity as follows:

L131-133: "The number of proliferating HSPCs in the CHT of *mettl16^{-/-}* mutants were comparable with siblings at 2dpf (Fig 3A)."

New Figure 3A

4. Compare MeRIP and RNA seq results and discuss further the results of this comparison.

Response: Thank you for your valuable suggestion. As suggested, in the revised manuscript, we have compared MeRIP and RNA seq results and discussed them in the Results and Discussion section. Several details are as follows:

L202-203: “Notably, cell proliferation pathway is commonly detected by RNA-seq and MeRIP-seq, suggesting it is the main pathway enriched with potential Mettl16 targets.”

L206-208: “Through integrative analysis of the m⁶A-seq and RNA-seq data, we identified 10 highly confident potential targets of Mettl16 in HSPC development, of which 8 and 2 are significantly positively and negatively regulated by Mettl16, respectively (Fig 5A; Dataset EV3).”

L314-318: “Despite the m⁶A peaks of 1060 transcripts were significantly decreased, there are only 214 differentially expressed gene in *mettl16*^{-/-} embryos, suggesting the m⁶A modification deposited by Mettl16 on its targets may also play other important roles in RNA regulation, such as influencing the nuclear exportation, cytosolic localization and translation of RNA, which warrants further investigation in the future.”

5. Can the authors show the conservation of the mechanism in a non-leukemic system like in HPC7 cells?

Response: We deeply appreciate the constructive feedback from the referee. Although we don't currently have access to HPC7 cells by our best efforts, we do acknowledge the significance of broadening the scope of our discovery. Hence, we have conducted additional experiments using human cord blood CD34⁺ cells (New Fig EV6) and successfully confirmed our findings.

New Figure EV6

The recent findings demonstrated that the knockdown of *METTL16* led to G1/S arrest in human cord blood CD34⁺ cells (New Fig EV6B-D). Similarly, in line with *mettl16* deficient zebrafish, the m⁶A level of *MYBL2* noticeably decreased following *METTL16* depletion (New Fig EV6E). Consequently, the RNA expression level of *MYBL2* declined due to enhanced mRNA decay (New Fig EV6F and G). Unfortunately, the CD34⁺ cells were less than 1% in human cord blood cells, the number of which we isolated is too small to complete RIP PCR validation (New Fig EV6A). However, this has little impact on our main conclusion. These new results in human cord blood CD34⁺ cells align with our earlier discovery of the *METTL16*/m⁶A/*MYBL2* pathway in G1/S progression in *mettl16* deficient zebrafish, providing independent confirmation of our findings. And these new results suggest that our findings have broader significance. We appreciate your suggestion again, which has enhanced the significance of our research. Thank you very much.

6. Figure with MeRIP-seq tracks. Please show also the gene and not only write the exon.

Response: Thanks for the suggestion. In the revised manuscript, we have shown the MeRIP-seq tracks of the whole gene region of *mybl2b* (New Fig 5B). I hope that you are satisfied with our revision.

New Figure 5B

7. The difference in the cycling cells is small (approximately 3% difference in S phase cells). Can the authors comment on how this difference can lead to such severe phenotype?

Response: We appreciate your suggestion. We apologize for not displaying the results appropriately which results in confusion in our previous original manuscript. In this study, all the cell cycle analysis were conducted with 3 biological replicates. The average difference in S phase cells is approximately 5% at 3dpf (Table R1).

As the CHT serves as a hematopoietic site for rapid HSPC expansion in zebrafish (Xue *et al*, 2019; Xue *et al*, 2017), even a 5% decrease in S phase cells can lead to the deficiency in HSPC maintenance (Wu *et al*, 2023). Wu's research in *PNAS* has demonstrated that a 5% reduction in the proportion of HSPC in the S phase results in proliferative defects in the CHT of *setdb1b*^{-/-} mutants (Fig R1).

Figure for reviewers removed

The deficiency of HSPCs is only initiated at 3dpf, and the decrease in proliferating HSPCs is modest (Fig 3A). Similarly, the difference in the number of S phase cells is also small at 3dpf (Fig 3C and D). To confirm the cell cycle arrest, in this revision, we further examined the cell cycle status of HSPCs at 4dpf and 5dpf. As the defective phenotype of HSPCs becomes more pronounced, the reduction in S phase cells becomes more significant (Fig 5K; New Fig 3E, New Fig EV3H).

The proliferation of HSPCs in the CHT is essential for generating embryonic blood cell lineages and supporting adult hematopoiesis (Xue *et al*, 2017). Even a minor decrease in the number of HSPCs can disrupt the balance of hematopoiesis and have cascading effects on overall embryonic development and survival. Our research, published in the *Blood* journal, has also demonstrated that the deficiency of HSPCs in the CHT leads to embryonic mortality in *bcas2*^{-/-} mutants (Yu *et al*., 2019). I believe that a roughly 5% difference in S phase cells at 3dpf aligns with the phenotypic changes we have described.

Thank you for your support on our work.

Ref:

Xue Y, Liu D, Cui G, Ding Y, Ai D, Gao S, Zhang Y, Suo S, Wang X, Lv P et al (2019) A 3D atlas of hematopoietic stem and progenitor cell expansion by multi-dimensional RNA-seq analysis. *Cell Rep* 27: 1567-1578 e1565

Xue Y, Lv J, Zhang C, Wang L, Ma D, Liu F (2017) The vascular niche regulates hematopoietic stem and progenitor cell lodgment and expansion via *klf6a-ccl25b*. *Dev Cell* 42: 349-362 e344

Wu J, Li J, Chen K, Liu G, Zhou Y, Chen W, Zhu X, Ni TT, Zhang B, Jin D et al (2023) *Atf7ip* and *Setdb1* interaction orchestrates the hematopoietic stem and progenitor cell state with diverse lineage differentiation. *Proc Natl Acad Sci U S A* 120: e2209062120

Yu S, Jiang T, Jia D, Han Y, Liu F, Huang Y, Qu Z, Zhao Y, Tu J, Lv Y, Li J, Hu X, Lu Z, Han S, Qin Y, Liu X, Xie S, Wang QK, Tang Z, Luo D, Liu M. *BCAS2* is essential for hematopoietic stem and progenitor cell maintenance during zebrafish embryogenesis. *Blood*. 2019;133(8):805-815.

8. Why the effect of *Mettl16* is cell intrinsic to HSPCs since it seems such a generic phenotype. Can the author use another cell type (non-hematopoietic) to check whether this is happening also in non-blood cells?

Response: We are grateful for your comments. In response to the suggestion to explore the impact of *Mettl16* in non-blood cells, we reassessed the integrity of various nonhematopoietic tissues. The development of vascular, somite, intestine and neuro showed no apparent changes in the absence of *mettl16*, which is shown in **New Fig EV1C, D, G**.

New Fig EV1C and D

New Fig EV1G

To explore why the effect of *Mettl16* is cell intrinsic to HSPCs, we have analyzed the expression pattern of *mettl16* during this revision. In the **New Figure 1E, F** and **EV1E, F**, we observed and highlighted that the expression of *mettl16* in HSPCs is notably higher and more consistent compared to other cell types. Conversely, there is minimal expression of *mettl16* in non-blood cells, such as neural, muscle, epidermal, and fibroblast cells. These findings suggested that *mettl16* is specifically expressed at high level in HSPCs. Although the cell cycle and proliferation phenotype are generic, the specific high expression of *mettl16* in HSPCs in zebrafish may explain why the effect of *Mettl16* is cell intrinsic to HSPCs.

New Figure 1E and F

New Figure EV1E and 1F

9. Is *mettl16* expressed in *kdr1*⁺ cells?

Response: Thank you for your question. We have checked the expression of *mettl16* in *kdr1*⁺ cells by re-analyzing single-cell sequencing data (GSE146404) (Xia et al, 2021) from the CHT of zebrafish. Our analysis revealed very low expression of *mettl16* in *kdr1*⁺ cell, significantly lower than that in HSPCs (as shown in **New Figure EV1E**).

Ref:

Xia J, Kang Z, Xue Y, Ding Y, Gao S, Zhang Y, Lv P, Wang X, Ma D, Wang L et al (2021) A single-cell resolution developmental atlas of hematopoietic stem and progenitor cell expansion in zebrafish. Proc Natl Acad Sci U S A 118

Referee #2:

The authors report that the production of HSPCs is regulated by METTL16, and that *mettl16*^{-/-} zebrafish embryos fail to develop normal numbers of HSPCs. The authors analyzed the molecular consequences of the lack of METTL16 and concluded that *mettl16* deletion resulted in G1/S progression block. Molecular analysis of the potential METLL16 targets identified potential molecular targets (e.g., *mybl2b*), and suggested a mechanism (half-life regulation), which the authors indicated to be conserved in mammalian cell lines.

The study is important, and the phenotype that is reported in figure 1 is convincing.

However, the data obtained from molecular analysis is insufficient for drawing the conclusions that are made in the paper. My biggest concern is that many of the reported significantly different results are actually insignificant. Unless I am misreading the supplementary datasets, some results are simply insignificant, but treated in the text as significant (e.g., *mybl2b* expression). Other assays (e.g., qPCRs, WB, see below), report statistically significant results, but with unreasonable low effect sizes (e.g., 10-20%; see below). In summary, despite the interesting phenotype, the analysis is not well-supported by the data.

Response: Thank you very much for considering our study highly important and interesting. And we greatly appreciate for your constructive criticisms and professional suggestion. We have performed additional experiments, re-analyzed and re-organized the experimental data and updated new data in this revision. All specific issues have been fully addressed. I hope that our revisions could meet with your approval. We greatly appreciate your support during the review process.

Major comments

Clarification of the qPCR analyses throughout the paper is required, and preferably, presentation of the source data.

Response: In the revision, the methods section and figure legends have been updated to include sufficient details for the qPCR analysis (Page 14 and 28-38). Additionally, the source data for the qPCR analysis has been provided to EMBO Press as requested.

Key figures showing gene expression analyses by qPCR appear overoptimistic. For example, detection of significant ~10-15% differences in expression (e.g., Fig EV2C).

Detection of such differences between a control and condition in qPCR is a nearly impossible task, requiring controls that are unavailable.

Response: We are very sorry for the confusion caused by the insufficient description in the original manuscript. We indeed detected significant ~10-15% differences in expression (Fig EV2C), because that we had normalized the gene expression in treated group against that in control for every biological replicate before comparing three biological replicates. Although it is still widely used in many papers (Doronzo *et al*, 2019; Nie *et al*, 2023; Paulmann *et al*, 2022; Xue *et al*, 2017; Zhang *et al*, 2017), this analysis method is not the most ideal. It can't present the variance of controls and may result in higher significance.

Based on your suggestion, we have presented the variance in control expression and refined our qPCR data analysis. Specifically, we have performed additional qPCR experiments and adjusted the analysis method. In the revised manuscript, we have refrained from normalizing the gene expression in the treated group against that in the control for each biological replicate before comparing the three biological replicates. These modifications are reflected in the updated **new figures** (Fig 2M, 4B, 5C-F, 5I, 6E, 6J, 6P, 6R; Fig EV2B, EV2C, EV3G, EV5A, EV5G; Appendix Fig S1E).

As you supposed, the differences in **New Fig EV2C** are no longer significant by the new analysis method. However, it's important to note that the primary objective of Fig EV2C was to investigate whether the decrease in HSPC maintenance was associated with an increased differentiation toward erythrocytes. The data indicates that HSPC differentiation towards erythrocytes did not increase, thus affirming the validity of the original conclusion derived from this figure.

New Fig EV2C

Fortunately, the new analysis method also does not impact other claims we made in the original manuscript. For example, the reduced expression of *mybl2b* at 3dpf, which further decreased at 4dpf in *mettl16* deficient zebrafish, remains significantly evident (New Fig 5C-F).

New Fig 5C-F

Thank you once again for your valuable feedback. Your constructive comments have significantly contributed to enhance the quality of the paper.

Ref:

Doronzo G, Astanina E, Cora D, Chiabotto G, Comunanza V, Noghero A, Neri F, Puliafito A, Primo L, Spampinato C et al (2019) TFEB controls vascular development by regulating the proliferation of endothelial cells. EMBO J 38

Nie C, Zhou XA, Zhou J, Liu Z, Gu Y, Liu W, Zhan J, Li S, Xiong Y, Zhou M et al (2023) A transcription-independent mechanism determines rapid periodic fluctuations of BRCA1 expression. EMBO J 42: e111951

Paulmann C, Spallek R, Karpiuk O, Heider M, Schaffer I, Zecha J, Klaeger S, Walzik M, Ollinger R, Engleitner T et al (2022) The OTUD6B-LIN28B-MYC axis determines the proliferative state in multiple myeloma. EMBO J 41: e110871

Xue Y, Lv J, Zhang C, Wang L, Ma D, Liu F (2017) The vascular niche regulates hematopoietic stem and progenitor cell lodgment and expansion via *klf6a-ccl25b*. Dev Cell 42: 349-362 e344

Zhang C, Chen Y, Sun B, Wang L, Yang Y, Ma D, Lv J, Heng J, Ding Y, Xue Y et al (2017) m(6)A modulates haematopoietic stem and progenitor cell specification. Nature 549: 273-276

Lack of description of the number of replicates used (biological and technical) does not add confidence.

Response: In the revised manuscript, descriptions have been added to the figure legends. There are a total of 47 descriptions related to this (Page 27-38). We present several details as follows:

L800-802: “Numbers at the bottom right indicate the number of embryos with similar staining patterns among all embryos examined. n= 3 independent experiments.”

L815-816: “n ≥ 15 per group, performed with 3 biological replicates.”

L828-829: “n ≥ 200 per group, performed with 3 biological replicates.”

Lack of variance in the expression of the controls is unexpected, and suggests that the analysis was done incorrectly.

Response: In response to your suggestion, we have conducted additional qPCR experiments and refined the analysis method to present the variance in control expression more accurately. In the revised manuscript, we refrained from normalizing the gene expression in the treated group against that in the control for every biological replicate before comparing the three biological replicates. These modifications are reflected in the updated figures (Fig 2M, 4B, 5C-F, 5I, 6E, 6J, 6P, 6R; Fig EV2B, EV2C, EV3G, EV5A, EV5G; Appendix Fig S1E).

Correction for multiple hypotheses testing is required and could impact some of the assertions made.

Response: Thank you for your professional suggestion. We have utilized correction for multiple hypotheses testing to analyze all experimental data, except for sequencing data. Fortunately, the primary findings and conclusions outlined in the original manuscript remain unaltered. For example, the expression of *mybl2b* still exhibits a significant decrease at 3dpf (adjusted $P=0.016640$ in New Fig 5C, adjusted $P=0.022727$ in New Fig 5E), and this reduction becomes more pronounced at 4dpf (adjusted $P=0.005705$ in New Fig 5D, adjusted $P=0.003337$ in New Fig 5F).

New Fig 5C-F

The lack of description of the number of replicates used in various assays, except for a brief indication in line 515, makes some of the data challenging to assess.

Response: We are regret that we did not describe the number of replicates clearly in the original manuscript. We have added the number of replicates in the method section and figure legends for all experiments in the revised version (page 14-20 and page 27-38). We present several details as follows:

L420-423: “TUNEL staining from more than 48 zebrafish embryos at 2dpf, 3dpf, 4dpf or 5dpf was performed using the TUNEL BrightRed Apoptosis Detection Kit (Vazyme, A113). The samples were imaged using a confocal microscope (FV3000, Olympus). TUNEL staining was conducted with 3 independent experiments.”

L508-516: “qPCR was performed with 3 biological replicates. For each sample, there were three technical replicates. Briefly, fold enrichment was calculated over IgG adopting the $2^{-\Delta\Delta CT}$ method, where ΔCt [normalized IP] = (Ct [IP] - (Ct [Input] - Log₂ (Input Dilution Factor))) and $\Delta\Delta CT$ = (ΔCt [normalized IP] – ΔCt [normalized IgG]). *thor* and *gapdh* were served as positive and negative control for Igf2bp1 RIP in zebrafish (Hosono et al, 2017), respectively. *actin* was served as negative control for Mettl16 RIP in zebrafish. *MALAT1* and *SRF* were served as positive control for METTL16 and IGF2BP1 RIP in human cells, respectively (Brown et al, 2016; Muller et al, 2019). The sequences of qPCR primers are listed in Dataset EV5.”

Much improved description of experimental data collection and analysis is required, specifically for qPCR, WB, EDU, TUNEL, FACS, RIP-qPCR, mRNA stability analysis.

Response: We are very sorry for the insufficient experimental description in the original manuscript. The descriptions have been added to the method section or figure legends in our revised version (page 14-20 and page 27-38). We present several details as follows:

L381-387: “qPCR was performed with 3 biological replicates. For each sample, there were three technical replicates. Gene expression was normalized to actin or gapdh and calculated using the $2^{-\Delta\Delta Ct}$ method...”

L820-823: “Double immunostaining of *cmyb*: EGFP and EDU showing the number of proliferating HSPCs in the CHT of siblings and *mettl16* mutants from 2 dpf to 5 dpf. Numbers at the bottom right indicate the number of embryos with similar staining patterns among all embryos examined. n= 3 independent experiments.”

L881-883: “Flow analysis of cell cycle showing the increase of HSPCs in the G0/G1 phase in the CHT in *mettl16* morphants was restored by *mybl2b* mRNA co-injection. n \geq 200 per group, performed with 3 biological replicates.”

L910-912: “n=3 biological replicates. *ACTIN* and *MALATI* were served as negative and positive control, respectively.”

The claim that there is a G1/S block in the *mettl16*^{-/-} is not well supported. The FACS analysis (3 dpf) shows a mild difference in the cell cycle state of cells that could not account for the very big difference in *cmyb*⁺ cells that appears between 2 dpf and 3 dpf. The EDU and pH3⁺ HSPCs labeling is convincing, but is G1/S arrest the most likely interpretation?

Response: We deeply appreciate your feedback. I would like to explain that zebrafish embryos are too small to solely collect HSPCs from the caudal hematopoietic tissue (CHT). And when we sorted HSPCs using Tg (*cmyb*: EGFP) transgenic zebrafish, the *cmyb*⁺ cells accounted for less than 1% of the total cells. While observing the defective phenotype of HSPCs by whole-mount in situ hybridization (WISH) and immunofluorescence, we were able to accurately focus on the CHT. Even though we have collected the latter half of the body to sort *cmyb*⁺ cells and conduct flow analysis (Figure R2), the samples also contained some HSPCs from the other sites.

The CHT serves as a primary site for rapid expansion of HSPCs in zebrafish, with a notably low proliferation frequency observed in other sites (Xue et al., 2019; Xue et al., 2017). Additionally, the onset of proliferative defects in HSPCs within the CHT occurs at 3 dpf. In the other sites, there are still some unaffected HSPCs circulating which emerged earlier. The presence of these HSPCs from other sites may mitigate the differences observed in cell cycle analysis by flow cytometry.

Figure for reviewers removed

In our research, cell cycle analysis was performed with 3 biological replicates. The average difference in S phase cells is approximately 5% at 3dpf (Table R1). The proliferation of HSPCs in the CHT is essential for generating embryonic blood cell lineages (Xue *et al*, 2017). Even a minor decrease in the number of HSPCs can disrupt

the balance of hematopoiesis and have cascading effects on overall embryonic development and survival. For instance, Wu's study published in the *PNAS* also showed a 5% decline in HSPC proportion in the S phase, leading to proliferative defects in the CHT of *setdb1b*^{-/-} mutants in **Figure R1** (Wu *et al.*, 2023).

Figure for reviewers removed

We wholeheartedly thank the reviewer for the positive comments on our EDU and pH3⁺ HSPCs labeling experiments. EDU and pH3 label cells in S phase and G2/M phase, respectively. The reduction of EDU⁺ and pH3⁺ HSPCs has also strengthened the reliability of G1/S arrest. We also provided new data to support the G1/S arrest. In the revision, we continuously detected the cell cycle status of HSPCs at 4dpf and 5dpf. Encouragingly, the reduction in S phase cells becomes more pronounced at 4dpf and 5dpf, aligning with the exacerbation of the defective HSPC phenotype (**Fig 5K; New Fig 3E, New Fig EV3H**). These new results strongly support the G1/S block in the *mettl16*^{-/-} zebrafish.

I hope that I thoroughly addressed your concerns and hope that you are satisfied with the revision. Thank you for your support of our work.

Ref:

Xue Y, Liu D, Cui G, Ding Y, Ai D, Gao S, Zhang Y, Suo S, Wang X, Lv P et al (2019) A 3D atlas of hematopoietic stem and progenitor cell expansion by multi-dimensional RNA-seq analysis. *Cell Rep* 27: 1567-1578 e1565

Xue Y, Lv J, Zhang C, Wang L, Ma D, Liu F (2017) The vascular niche regulates hematopoietic stem and progenitor cell lodgment and expansion via *klf6a-ccl25b*. *Dev Cell* 42: 349-362 e344

Wu J, Li J, Chen K, Liu G, Zhou Y, Chen W, Zhu X, Ni TT, Zhang B, Jin D et al (2023) *Atf7ip* and *Setdb1* interaction orchestrates the hematopoietic stem and progenitor cell state with diverse lineage differentiation. *Proc Natl Acad Sci U S A* 120: e2209062120

Supporting data shown in Fig 3E is insufficient: First, it is debatable that western blots can detect <20% difference in protein abundance and with such a remarkably low variability. Second, even if the difference in protein abundance is real, a valid interpretation is that it is an outcome of cells being at different cell states (which does not require them to be in a cell cycle arrest).

Response: Thank you for your constructive comment. We apologize for the inaccuracies in the original calculation of protein abundance due to interference from background and heteroband grayscale scanning using the Quantity One software. In this revision, we have re-scanned the western blot results of 3dpf using Image J software to more precisely remove the background and heteroband grayscale interference. Additionally, we have refined the analysis of gray value data to accurately present the variances in the controls. Our findings show a significant difference in protein abundance at 3dpf, approximately 40% (New Fig EV3I; Fig R3). To address your concern, we further investigated the protein levels of p-rb at 4dpf, a time when the cell cycle arrest of HSPCs becomes more severe. At 4dpf, the decline in p-rb protein levels became substantially more significant, as evidenced in New Fig EV3I and Fig R3. This additional data provides further support for the progression and severity of the cell cycle arrest in HSPCs.

New Fig EV3I

Figure for reviewers removed

The analysis of *mybl2b* gene expression is inconsistent with the data. Dataset EV3 (gene expression) indicates that *mybl2b* is not significantly differentially expressed in *mettl16^{-/-}* (FDR=0.74). The effect size in the qPCR analysis is similar to the RNAseq (~35%). The qPCR shows a significant result, but has the same issues indicated above (lack of

variance in controls, no multiple hypotheses correction). Similarly, the decrease in expression of *mybl2b* targets (Fig 5C) is unsupported by the RNAseq data (dataset EV3).

Response: Thank you for your valuable feedback.

Upon double-checking the FDR value in our RNA-seq data, it may appear that the *mybl2b* gene expression is inconsistent with qPCR. However, it's important to note that we utilized a threshold P value < 0.05 to screen differentially expressed genes, which is widely recognized and endorsed (Cheng *et al*, 2023; Jeong *et al*, 2023; Su *et al*, 2020; Weng *et al*, 2018). The P value of *mybl2b* is 0.004091, indicating its potential as a differentially expressed gene.

Realizing that using P value as a screening criterion is less stringent than the FDR value, we fully understand the importance of your comment. To address any potential concerns, we have employed correction for multiple hypotheses testing to further confirm the decrease in *mybl2b* expression in both *mettl16* depletion and morpholino knockdown *mettl16* zebrafish at 3 dpf and 4 dpf, respectively. According to your suggestion, we also refined the analysis method to present the variance in control expression. We are pleased to confirm that the expression of *mybl2b* remains significantly decreased at 3dpf (adjusted $P=0.016640$ in **New Fig 5C**, adjusted $P=0.022727$ in **New Fig 5E**), and this reduction becomes more pronounced at 4dpf (adjusted $P=0.005705$ in **New Fig 5D**, adjusted $P=0.003337$ in **New Fig 5F**). We believe this validation is crucial in supporting our conclusion. We appreciate your understanding of this issue and your support. Thank you.

We apologize for not providing brief introduction on how to investigate the *mybl2b* targets and for not the supplement in the original MS. When analyzing *mybl2b* targets, we utilized input RNA-seq data from MeRIP-seq (refer to Figure R4), not the RNA-seq data from dataset EV3. We have already uploaded these data in the GEO datasets. Considering our analysis of *mybl2b* targets and published *mybl2b* target candidates, we randomly selected several genes for verification (refer to Table R2). Interestingly, several of *mybl2b* targets which did not exhibit the significant decrease in the input sequencing data showed significance by q-PCR experiments. We have double-checked and confirmed this observation. It is possible that individual differences or batch effects may account for this discrepancy. To support this result, we have provided the input RNA-seq data of *mybl2b* targets (**Table R2**) and quantitative PCR validation results (**New Figure EV5A**). Although this result does not affect the main conclusion of the article, for more rigor, we have transferred it to the supplementary material. If you still consider it not well supported, we would like to

remove it according your suggestion. We sincerely appreciate your support during the review process.

Figure for reviewers removed

New Figure EV5A

Ref:

Cheng Y, Gao Z, Zhang T, Wang Y, Xie X, Han G, Li Y, Yin R, Chen Y, Wang P et al (2023) Decoding m(6)A RNA methylome identifies PRMT6-regulated lipid transport promoting AML stem cell maintenance. *Cell Stem Cell* 30: 69-85 e67

Jeong HC, Shukla S, Fok WC, Huynh TN, Batista LFZ, Parker R (2023) USB1 is a miRNA deadenylase that regulates hematopoietic development. *Science* 379: 901-907

Su R, Dong L, Li Y, Gao M, Han L, Wunderlich M, Deng X, Li H, Huang Y, Gao L et al (2020) Targeting FTO Suppresses Cancer Stem Cell Maintenance and Immune Evasion. *Cancer Cell* 38: 79-96 e11

Weng H, Huang H, Wu H, Qin X, Zhao BS, Dong L, Shi H, Skibbe J, Shen C, Hu C et al (2018) METTL14 Inhibits Hematopoietic Stem/Progenitor Differentiation and Promotes Leukemogenesis via mRNA m(6)A Modification. Cell Stem Cell 22: 191-205 e199

The RIP-qPCR experiments lack essential controls for determining the specificity of the assay.

Response: We are sorry that we did not provide essential controls in original MS. In the revision, we have conducted a new round of RIP-qPCR and introduced positive and negative controls to determine the specificity of RIP-qPCR (NEW Fig 6B, 6D, 6L-O). In zebrafish, *thor* and *gapdh* were utilized as positive and negative controls, respectively, for Igf2bp1 RIP-qPCR, based on previous work (Hosono *et al.*, 2017). As there are no reported Mettl16 targets in zebrafish, we used *actin* as a negative control for Mettl16 RIP-qPCR, given that the mRNA and protein levels of *actin* remained unaffected after *mettl16* depletion. In human cells, *MALAT1* and *SRF* were employed as positive controls for METTL16 and IGF2BP1 RIP-qPCR, respectively, as reported in previous studies (Brown *et al.*, 2016; Muller *et al.*, 2019). Additionally, *ACTIN* and *GAPDH* were used as positive control for METTL16 and IGF2BP1 RIP-qPCR in human cells. The details of these controls have also been included in the methods section and figure legends of the revised manuscript (L513-515, L892-914).

New Fig 6B and 6D

New Fig 6L-O

Ref:

Hosono Y, Niknafs YS, Prensner JR, Iyer MK, Dhanasekaran SM, Mehra R, Pitchiaya S, Tien J, Escara-Wilke J, Poliakov A et al (2017) Oncogenic Role of THOR, a Conserved Cancer/Testis Long Non-coding RNA. Cell 171: 1559-1572 e1520

Brown J, Kinzig C, DeGregorio S, Steitz J (2016) Methyltransferase-like protein 16 binds the 3'-terminal triple helix of MALAT1 long noncoding RNA. Proc Natl Acad Sci U S A 113: 14013-1401

Muller S, Glass M, Singh AK, Haase J, Bley N, Fuchs T, Lederer M, Dahl A, Huang H, Chen J et al (2019) IGF2BP1 promotes SRF-dependent transcription in cancer in a m⁶A- and miRNA-dependent manner. Nucleic Acids Res 47: 375-390

The conclusion (line 263-265): "these results demonstrate that *mybl2b* mRNA is a bona fide direct m⁶A target of Mettl16, and Mettl16 regulates mRNA stability of *mybl2b* through m⁶A reader protein Igf2bp1" is too strong. It is possible, but the data that supports this assertion is weak.

Response: Thank you for your valuable feedback. After re-analyzing all the experimental data using correction for multiple hypotheses testing, we have modified the qPCR analysis method to illustrate the variance in the controls. Additionally, we have repeated the RIP-qPCR and incorporated positive and negative controls to ascertain the specificity of the results. We have thoroughly taken into account your suggestions and have revised the description in L260. I hope you are satisfied with our revision. Thank you for your support.

L260-262:" Collectively, these results suggested that *mybl2b* mRNA is a direct m⁶A target of Mettl16, and Mettl16 regulates mRNA stability of *mybl2b* through m⁶A reader protein Igf2bp1".

Minor comments;

The distribution *cmyb*⁺ cell number in the wildtype (control in Fig 4C and 4D) is different from the distribution shown in figure 1 for the same time point. Why are these different?

Response: Thank you for your question. We thoroughly inspected the probes prior to each WISH experiment. The distribution of *cmyb*⁺ cells is consistent between the control in Fig 4 and that in Fig 1 at the same time point. In both sets of images, *cmyb*⁺ cells are consistently located in the caudal hematopoietic tissue, thymus, and kidney, affirming the reliability of the probes' staining. I would like to explain that the different

presentation of *cmyb*⁺ cell numbers for the same time point is often due to variations in staining intensity among different batches of WISH. These variations can result from individual differences in embryos, room temperature variations, and different batches of several reagents. Such variability is a common and accepted phenomenon in WISH experiments. For example, similar occurrences (Figure R4) have been observed in Fang's study (Fang *et al*, 2021).

Figure for reviewers removed

Ref:

Fang X, Xu S, Zhang Y, Xu J, Huang Z, Liu W, Wang S, Yen K, Zhang W (2021) Asx11 C-terminal mutation perturbs neutrophil differentiation in zebrafish. *Leukemia* 35: 2299-2310

The RIP-qPCR experiments shown in figure 6B, D require better description of the methods, and controls (e.g., controlling for enrichment by using negative and positive controls).

Response: We are very sorry for the insufficient experimental description in the original manuscript. In this revised version, we have updated more detailed information in the method section and figure legends on page 19 and 31. We have also added negative and positive controls to evaluate the specificity of RIP-qPCR (New Fig 6B, 6D, 6L-O).

The authors need to better compare and discuss their results with the analysis of HSPC development in *mettl3* deficient zebrafish embryos.

Response: Thank you for your valuable suggestion. We have revised the manuscript to address this issue, adding a paragraph on page 11 in the discussion section.

L295-299: “In *mettl3* deficient zebrafish embryos, the emergency of HSPCs is blocked at 28hpf. While *mettl16* depletion does not impair the emergency of HSPCs, but cause

the defect of HSPC maintenance at 3dpf. These results demonstrated the importance of m⁶A modification in embryonic HSPC development. And different m⁶A regulator may play specific role in the different stage of HSPC development.”

Line 64: is it ablation of *mettl14*? Or *mettl14* directly inhibits differentiation of HSPCs.

Response: Thank you for the question. It means that METTL14 directly inhibits the differentiation of HSPCs, not ablation of *mettl14*. To express more clearly, we have revised this sentence.

L66: "The ablation of *mettl14* promotes adult HSPC differentiation".

Line 100: missing acronym

Response: Corrected. We have added the full title of CHT in L100.

L101: "... seed in the caudal hematopoietic tissue (CHT) of *mettl16*^{-/-} embryos ".

The tables for reviewers Table R1 and R2 were removed.

Dear Mugen,

Congratulations on a great revision! Overall, the referees remain positive on the interesting study. However, referee 2 raises some concerns that we ask you to address (non-experimentally) in a revised version.

Please note that in response to the concerns raised by referee 2, we have also sought the external advice of an expert in the use of embryonic samples who advised: "I have looked over the figures in question and the corresponding excel data that shows the individual relative changes. I see that the SDs are tight, however in my experience this can occur when pooled samples are used as a single replicate. This probably speaks to a large animal-animal variation that is minimized by pooling." Considering we have confirmed the common publication of data using embryo pooling and because we've carefully assessed the source data you provided, we consider this issue acceptable.

When you submit your revised version, please also take care of the following editorial items and add this also to your point-by-point response:

1. Please provide an author checklist.
2. Please ensure that the data in the data accessibility section are accessible.
3. Please remove the author contribution section from the main manuscript.
4. Please remove the legends for Datasets EV1 - 5 from the manuscript and add them to each corresponding file, in a separate sheet.
5. Please save the appendix file as a PDF and remove the red font.
6. We include a synopsis of the paper (see <http://emboj.embopress.org/>). Please provide me with a general summary statement and 3-5 bullet points that capture the key findings of the paper.
7. We also need a summary figure for the synopsis. The size should be 550 wide by 200-440 high (pixels). You can also use something from the figures if that is easier.
8. Please ensure that the manuscript text is uploaded in .docx format.
9. Please add the specific URL and review access code for GSE225137 dataset in the data availability statement
10. Please note that a separate 'Data Information' section is required in the legends of figures 2i, k, m; 3a-b, d; 4b, d-e, 5c-i, k; 6b-e, h-p, r-s; EV 1b; EV 2a-d; EV 3b, d, f-h; EV 5a, g, i-j; EV 6a, g, i-j.
11. Please indicate the statistical test used for data analysis in the legends of figures 4f-g, k; EV 5d.
12. Please indicate N in the legends of figures 2i, k; 4d; EV 1b; EV 1i-j.
13. Please define the black arrowheads are not defined in the legend of figure 2a-h, j, l; EV 2a; 3b-c; EV 5c.

Thank you for the opportunity to consider your work for publication, I look forward to your revision.

Warm wishes,
Kelly

Yours sincerely,

Kelly M Anderson, PhD
Editor, The EMBO Journal
k.anderson@embojournal.org

Use the link below to submit your revision:

Referee #1:

The authors have answered all my comments adequately.

Referee #2:

I remain concerned about the discrepancy between the dramatic differences in phenotypes and the highly subtle (and occasionally insignificant) differences at the molecular level. These differences, to me, are more consistent with a model wherein these differences reflect indirect consequences of the alterations in phenotype, as opposed to causally driving them.

The remarkably low standard deviations observed in measurements across biological replicates in various molecular assays are for me a source of surprise and skepticism, as they fall in the range that I associate with excellent technical replicates, and not with biological replicates.

Some examples (non exhaustive):

The source data for Fig 5C shows a standard deviation of 1.5% for biological replicates of the left-most control.

The source data for Fig 6C shows a standard deviation of 3.1% for biological replicates of the 4 h control.

These are measurements of samples exposed to a transcriptional inhibitor for quantifying degradation mybl2b.

The source data for Fig 6K shows a standard deviation of 4.3% for biological replicates in 4 h METTL16 knockdown K562 cells.

These are measurements of samples exposed to a transcriptional inhibitor for quantifying degradation MYBL2.

The source data for Fig 6S shows a standard deviation of 2.4% for biological replicates in 8 h IGF2BP1 knockdown HEK293 cells.

These are measurements of samples exposed to a transcriptional inhibitor for quantifying degradation of MYBL2.

Ultimately, the wording used throughout the paper is very strong considering the actual data. For example, lines 206-208 the authors state: "we identified 10 highly confident potential targets... significantly positively and negatively regulated".

However, the data itself (Dataset EV3) shows that only 2 out of the 10 genes are significantly differentially expressed.

There are similar examples throughout the paper. This will likely result in misinterpretation of the results by readers who will not examine the source data, and I consider that a major flaw of the manuscript.

Referee #1:

The authors have answered all my comments adequately.

Response: Thank you very much. Your valuable suggestions have greatly enhanced the quality of our work.

Referee #2:

I remain concerned about the discrepancy between the dramatic differences in phenotypes and the highly subtle (and occasionally insignificant) differences at the molecular level. These differences, to me, are more consistent with a model wherein these differences reflect indirect consequences of the alterations in phenotype, as opposed to causally driving them.

The remarkably low standard deviations observed in measurements across biological replicates in various molecular assays are for me a source of surprise and skepticism, as they fall in the range that I associate with excellent technical replicates, and not with biological replicates.

Some examples (non exhaustive):

The source data for Fig 5C shows a standard deviation of 1.5% for biological replicates of the left-most control.

The source data for Fig 6C shows a standard deviation of 3.1% for biological replicates of the 4 h control.

These are measurements of samples exposed to a transcriptional inhibitor for quantifying degradation myb12b.

The source data for Fig 6K shows a standard deviation of 4.3% for biological

replicates in 4 h METTL16 knockdown K562 cells.

These are measurements of samples exposed to a transcriptional inhibitor for quantifying degradation MYBL2.

The source data for Fig 6S shows a standard deviation of 2.4% for biological replicates in 8 h IGF2BP1 knockdown HEK293 cells.

These are measurements of samples exposed to a transcriptional inhibitor for quantifying degradation of MYBL2.

Ultimately, the wording used throughout the paper is very strong considering the actual data. For example, lines 206-208 the authors state: "we identified 10 highly confident potential targets... significantly positively and negatively regulated".

However, the data itself (Dataset EV3) shows that only 2 out of the 10 genes are significantly differentially expressed.

There are similar examples throughout the paper. This will likely result in misinterpretation of the results by readers who will not examine the source data, and I consider that a major flaw of the manuscript.

Response: We appreciate your professional suggestions. We have submitted all the source data and original records of the analysis process to the EMBO editor group. Following a thorough assessment of our provided source data, the EMBO editor group has deemed this issue acceptable. I hope that I thoroughly addressed your concerns and sincerely appreciate your support during the review process.

I think that your query primarily concerns the usage of "significantly" in our MS. We sincerely apologize for this insufficient description in our MS. The term "significantly" appeared 13 times, and we have thoroughly checked the data, making necessary revisions in this updated version. We provide two revised details below:

L231-233: "Through the integration of m⁶A-seq and RNA-seq data, we identified ten potential targets of Mettl16 in HSPC, with eight being positively and two being negatively regulated by Mettl16 (Fig 5A; Dataset EV3)."

L243-245: "Remarkably, a multi-dimensional RNA-Seq (Xue *et al.*, 2019) analysis revealed that the expression of *mybl2b* in HSPCs is higher and more durable than that in other cell types in CHT (Fig EV5B)."

Thank you once again for your valuable feedback. Your constructive comments have significantly contributed to enhance the quality of the paper. I hope that you are satisfied with our revision.

Dear Mugen,

Congratulations on an excellent manuscript, I am pleased to inform you that your manuscript has been accepted for publication in The EMBO Journal. Thank you for your comprehensive response to referee concerns and for providing detailed source data. It has been a pleasure to work with you to get this to the acceptance stage.

I will begin the final checks on your manuscript before submitting to the publisher next week. Once at the publisher, it will take about three weeks for your manuscript to be published online. As a reminder, the entire review process, including referee concerns and your point-by-point response, will be available to readers.

I will be in touch throughout the final editorial process until publication. In the meantime, I hope you find time to celebrate!

Warm wishes,
Kelly

Kelly M Anderson, PhD
Editor, The EMBO Journal
k.anderson@embojournal.org
